# Accounting for Heterogeneous Parameters in Decision-Focused Learning

## Abstract

Decision-focused learning (DFL) is a recent machine learning paradigm aimed at tackling predict-then-optimize problems, where the task is to predict the parameter values of a parametric optimization problem from features. Instead of maximizing predictive accuracy, DFL maximizes downstream decision quality, by training the model to avoid specifically those errors that most negatively impact decision-making. In this work, we systematically investigate an understudied aspect of DFL: that for different parameters, the way prediction errors affect downstream decision quality may differ, and that the conventional model architecture cannot account for these differences. We first formalize this phenomenon as *parameter heterogeneity* and provide a strict theoretical characterization of when it arises. We then show that adjusting for this varying sensitivity to prediction errors does not require intricate architectural designs. Instead, significantly better decision quality can often be achieved through simple architectural alterations to equip models with the ability to learn parameter-specific predictive mappings. To this end, we investigate three architectural adaptations to the predictive model, and apply parameter-permutation-based data augmentation scheme to enhance data efficiency. We extensively evaluate the impact of these changes across several dimensions, using linear and nonlinear predictive models, and optimization problems of different complexities. Our findings show that significant performance gains can be realized through such architectural alterations and data augmentation scheme, across different problem types.

## 1 Introduction

Decision-making processes are commonly modeled as constrained optimization problems, which are solved by optimizing an objective function, subject to constraints. However, in many applications, some of the problem's parameters are not known at solving time. For example, consider a knapsack problem where a subset of items must be selected to maximize total profit under a strict weight capacity. The actual values of the items may depend on uncertain future conditions, such as fluctuating market prices. Consequently, these item values must first be *predicted* from available contextual features before the optimal selection decision can be computed.

Such problems can be framed as *predict-then-optimize problems* (Mandi et al., 2024), consisting of a prediction and an optimization stage. In the prediction stage, a machine learning model is used to predict values for the unknown parameters (e.g., item values), where each individual parameter value has its own corresponding vector of contextual features (e.g., historical demands or market data). In the optimization stage, the resulting constrained optimization problem (e.g., knapsack problem) is solved. Because the solver's computed solution depends entirely on the predicted parameter values, the final decision quality hinges heavily on how the predictive model is trained. To this end, the following two training paradigms can be distinguished.

In *prediction-focused learning* (PFL), the predictive model is trained to maximize the accuracy of the predicted parameters, using traditional losses like the mean squared error or negative log-likelihood. In other words, PFL does not consider the downstream optimization problem during training. Though simple and efficient, this generally leads to suboptimal decision-making. This is because, while perfect predictions lead

to perfect decisions, different kinds of imperfect predictions affect downstream decision-making in different ways (Shah et al., 2024).

In *decision-focused learning* (DFL), the predictive model is instead trained with a loss function that directly reflects downstream decision quality. That is, DFL still predicts parameter values, but trains the model by integrating the optimization solver into the training loop. This allows the model to prioritize minimizing precisely those errors that would most negatively alter the solution of the optimization stage. As numerous works have shown, models trained with DFL generally lead to better decisions than models trained with PFL (Amos & Kolter, 2017; Donti et al., 2017; Elmachtoub & Grigas, 2022; Mandi et al., 2022; Mulamba et al., 2021; Pogančić et al., 2020; Wilder et al., 2019).

*Comment to reviewers: moved this paragraph upwards for improved structure.* However, an understudied aspect of predict-then-optimize problems is that prediction errors do not always affect downstream decision quality in the same way for all parameters. Returning to our knapsack example, where the item values are parameterized and must be predicted while the item weights are known: in this setting, the exact same error in the prediction of an item's value can affect the optimal decision differently depending on whether it is made on a high-weight or a low-weight item. For instance, overestimating the value of a heavy item might lead it to be selected at the expense of multiple lighter, more valuable items, reducing decision quality. Conversely, the same error made in the value prediction of a light item might not affect decision quality at all, because the item would be selected either way. In this example, the impact of prediction errors on downstream decision quality varies across the parameters. We refer to such parameters as heterogeneous.

*Comment to reviewers: moved this paragraph downwards for improved structure.* Despite the presence of such heterogeneous parameters in many optimization problems, the conventional architecture of the predictive model cannot account for this heterogeneity. The *de facto* standard in existing work is to deploy a single, shared predictive model across all parameters. Each parameter's value is produced independently through a separate invocation of this shared model, mapping each item's specific feature vector to its predicted value (Shah et al., 2022; 2024; Wilder et al., 2019; Mandi et al., 2022; Mulamba et al., 2021; Mandi & Guns, 2020; Ferber et al., 2020).[1] Consequently, existing approaches force the predictive model to treat all parameters equally. By relying on a shared architecture, these approaches fundamentally fail to account for parameter heterogeneity and the distinct ways in which prediction errors on different parameters affect downstream decision quality.

In this work, we show that this conventional reliance on a shared architecture is demonstrably suboptimal for problems containing heterogeneous parameters. We demonstrate that *even when all parameters depend on their features in exactly the same way*, superior decision quality can often be achieved by equipping the predictive model with the ability to learn parameter-specific mappings. In other words, these gains in decision quality stem not from heterogeneity in the data-generating process, but from heterogeneity in how prediction errors propagate through the downstream optimization problem. To systematically study this phenomenon and how to best account for it, we make the following contributions:

1. We formalize the phenomenon of parameter heterogeneity in predict-then-optimize problems, demonstrating that identical prediction errors can impact downstream decision quality differently depending on the parameter. We characterize the structural conditions under which parameters behave homogeneously and introduce a continuous degree-of-heterogeneity metric that quantifies deviations from homogeneity.

2. We investigate three parameter-aware architectural adaptations that enable predictive models to learn parameter-specific mappings. Additionally, we employ a parameter-permutation-based data augmentation scheme designed to preserve data efficiency in low-data or high-dimensional regimes.

3. We conduct an extensive experimental evaluation across multiple optimization domains (including linear, mixed-integer, and continuous non-linear problems). Our results demonstrate that accounting

---

[1]To the best of our knowledge, only the synthetic benchmark introduced in Elmachtoub & Grigas (2022) (and later adopted in Tang & Khalil (2022a)), trains a separate model for each parameter, because each parameter depends on its correlated features in a distinct way.

for parameter heterogeneity significantly improves decision quality, and validate the efficacy of the data augmentation scheme.

## 2 Background

The DFL paradigm aims to solve predict-then-optimize problems, which involve the prediction of unknown objective parameters $y$ of a parametric constrained optimization problem of the following form:

$$z^\star(y) = \arg\min_z \ f(z; y) \ \text{ s.t. } \ z \in \Omega \tag{1}$$

For any given parameter vector $y \in \mathbb{R}^n$, solving the optimization problem involves finding a solution $z^\star(y) \in \mathbb{R}^m$ in non-empty feasible space $\Omega$ that minimizes objective function $f(z; y)$.[2]

We assume that the unknown objective parameters $y = [y_1 \ y_2 \ \dots \ y_n]$ are correlated with known feature vectors $x = [x_1 \ x_2 \ \dots \ x_n]$ according to some distribution $P$, wherein each parameter $y_i \in \mathbb{R}$ depends on its features $x_i \in \mathbb{R}^f$ according to the same underlying mapping. Though distribution $P$ is not known, we assume access to a training set of examples $D = \{(x^{(i)}, y^{(i)})\}_{i=1}^K$, drawn from $P$, where each $x^{(i)}$ is a vector of feature vectors and each $y^{(i)}$ is the vector of corresponding parameters.

This training set is used to train a model $m_\omega$ with learnable weights $\omega$ – usually a linear regressor or a neural network – which makes predictions $\hat{y} = [m_\omega(x_1) \ m_\omega(x_2) \ \dots \ m_\omega(x_n)]$. With slight abuse of notation, we will simply denote this as $\hat{y} = m_\omega(x)$.

Unlike conventional regression, the objective in training is *not* to optimize the accuracy of $\hat{y}$. Rather, one aims to make predictions $\hat{y}$ that lead to good decisions $z^\star(\hat{y})$ with respect to the real parameters $y$. This is measured by the *regret*, which expresses the suboptimality of the made decisions $z^\star(\hat{y})$ with respect to the ground-truth parameters $y$.

$$Regret(\hat{y}, y) = f(z^\star(\hat{y}); y) - f(z^\star(y); y) \tag{2}$$

Thus, the aim is to minimize the *expected regret*:

$$\min_\omega \mathbb{E}_{x,y \sim P}[Regret(m_\omega(x), y)]. \tag{3}$$

In practice, since distribution $P$ is unknown, the expected regret is approximated using the training set $D$ via empirical risk minimization (ERM):

$$\min_\omega \ \frac{1}{K} \sum_{i=1}^K Regret\big(m_\omega(x^{(i)}), y^{(i)}\big). \tag{4}$$

Using this ERM framework, we next contextualize how existing DFL methodologies attempt to solve this problem.

## 3 Related work

The aim of DFL is to train predictive models that maximize downstream decision quality, rather than predictive accuracy. Methodologically, it is closely related to differentiable optimization, which offers a way of differentiating through any kind of continuous convex optimization problem (Amos & Kolter, 2017; Agrawal et al., 2019).

Most work in DFL, however, focuses on *combinatorial* optimization problems, which pose an additional challenge due to their discrete nature, making the gradient of the regret zero almost everywhere. Existing

---

[2] For notational simplicity and readability, we treat the optimal solution $z^*(y)$ as a unique vector throughout this work. In cases where multiple optimal solutions exist, our theoretical results continue to hold by assuming a deterministic tie-breaking rule that preserves symmetries present in the problem, or equivalently, by adopting a set-valued optimal solution map. However, we forego a fully set-valued exposition to avoid unnecessary technical complexity that does not alter our core findings.

work has circumvented this obstacle in different ways. A first approach is based on analytical smoothing of the optimization problem, leading to non-zero gradients of the regret (Wilder et al., 2019; Mandi & Guns, 2020). Another approach focuses on computing informative weight updates by perturbing the predicted parameters (Pogančić et al., 2020; Berthet et al., 2020; Niepert et al., 2021). Finally, surrogate loss functions can be used, like the seminal SPO+ loss (Elmachtoub & Grigas, 2022), various learnable losses (Shah et al., 2022; 2024; Bansal et al., 2024), losses based on noise-contrastive estimation (Mulamba et al., 2021) or learning to rank (Mandi et al., 2022), and losses based on the geometry of the feasible space (Tang & Khalil, 2024; Berden et al., 2025). For a comprehensive overview, we refer to (Mandi et al., 2024).

Another direction within DFL has instead focused on improving scalability. Because DFL involves repeatedly solving optimization problems, training can be significantly slower than in PFL. To improve scalability, prior work has investigated the use of problem relaxations and warm-starting techniques (Mandi et al., 2020), computationally cheaper solution caches in place of exact solvers (Mulamba et al., 2021), and surrogate losses that bypass expensive optimization entirely (Tang & Khalil, 2024; Berden et al., 2025).

In summary, existing DFL literature primarily focuses on engineering richer learning signals or improving algorithmic scalability. Our work shifts focus to a different dimension: parameter heterogeneity. We demonstrate that downstream decision quality is often sensitive to *where* prediction errors occur in the optimization problem. This asymmetry reveals an important limitation in conventional shared predictive models, and provides a motivation for parameter-aware model architectures.

## 4 Heterogeneous parameters

The central idea of this paper is that in predict-then-optimize problems, prediction errors do not always affect downstream decision quality in the same way for the different parameters. Before formalizing this and theoretically characterizing exactly when it happens, we give an illustrative example.

**Example 1.** Consider a parametric knapsack problem:

$$z^\star(y) = \arg\max \ y_1 z_1 + y_2 z_2 + y_3 z_3$$
$$\text{s.t.} \ \ z_1 + z_2 + 2z_3 \le 2$$
$$z_1, z_2, z_3 \in \{0, 1\}$$

In this problem, prediction errors on parameters $y_1$ and $y_3$ do not affect downstream decision-making in the same way, due to the differing coefficients associated with $z_1$ and $z_3$ in the constraint. We call $y_1$ and $y_3$ *heterogeneous* parameters. Take, for example, $y = [7\ 4\ 4]$ and $\hat{y} = [9\ 4\ 4]$. Even though value 7 is mispredicted as 9, parameter vector $\hat{y}$ still leads to correct decision $z^\star(\hat{y}) = [1\ 1\ 0]$, and thus to zero regret. However, if this same error were made on the third parameter instead (i.e., for $y = [4\ 4\ 7]$ and $\hat{y} = [4\ 4\ 9]$), then solution $z^\star(\hat{y}) = [0\ 0\ 1]$ would wrongfully put the third item in the knapsack, leading to a regret of 1.

On the other hand, prediction errors on parameters $y_1$ and $y_2$ affect downstream decision quality in the same way. We call them *homogeneous* parameters. For any $y$ and $\hat{y}$, the regret remains the same before and after interchanging $y_1$ and $y_2$ as well as $\hat{y}_1$ and $\hat{y}_2$.

We now formalize the difference between homogeneous and heterogeneous parameters as follows.

**Definition 1.** *Let $\sigma_{ij}$ be the transposition that interchanges the values of parameters $y_i$ and $y_j$. Then, parameters $y_i$ and $y_j$ are* homogeneous *if and only if, for all $\hat{y}$ and $y$, the regret remains invariant under transposition $\sigma_{ij}$, i.e., $Regret(\hat{y}, y) = Regret(\sigma_{ij}(\hat{y}), \sigma_{ij}(y))$. Parameters $y_i$ and $y_j$ are* heterogeneous *when they are not homogeneous.*

This definition expresses that for heterogeneous parameters, the impact of a parameter misprediction $\hat{y}_k$ is not only characterized by the *values* of $\hat{y}_k$ and $y_k$, but also by the value of $k$, i.e., by *which* parameter the prediction error is made on.

In Example 1, the homogeneous and heterogeneous parameters could be identified directly: homogeneous parameters corresponded to items with identical weights, whereas heterogeneous parameters corresponded to items with different weights. In general, however, this distinction is not always apparent from the optimization problem itself. Pathological cases may arise in which parameters appear heterogeneous but are actually homogeneous. For example, two items whose weights both exceed the knapsack capacity induce homogeneous objective parameters, even if their weights differ, since neither item can be selected in any feasible solution. This observation motivates the need for a precise understanding, which we now develop.

## 4.1 Conditions for homogeneity

Because parameter homogeneity is defined strictly in terms of regret, its evaluation depends exclusively on solutions that are optimal for some (true or predicted) parameter vector. Consequently, only the solutions that are optimal for *some* parameter vector play a role in our analysis. We refer to these as *supported* solutions.

**Definition 2.** *A feasible solution $z \in \Omega$ is* supported *if and only if there exists a parameter vector $y \in \mathbb{R}^n$ for which $z = z^\star(y)$.*

Before presenting our main theoretical result, we provide a supporting lemma that will be used in its proof. This lemma expresses that if two parameter vectors produce an identical optimal solution, then transposing their homogeneous parameters must preserve that equivalence. For a proof of the lemma, we refer the reader to Appendix A.

**Lemma 1.** *Let $y_i$ and $y_j$ be two homogeneous parameters. Then, there do not exist any two parameter vectors $y$ and $y'$ for which $z^\star(y) = z^\star(y')$ but $z^\star(\sigma_{ij}(y)) \neq z^\star(\sigma_{ij}(y'))$.*

We now arrive at our main theoretical result. Two parameters $y_i$ and $y_j$ are homogeneous if and only if, through a transformation $\tau$, the quality of every supported solution $z$, relative to the optimal solution, is maintained when interchanging $y_i$ and $y_j$. That is, objective function $f$ is invariant (modulo a constant) to the interchanging of values $y_i$ and $y_j$, and the simultaneous modification of solution $z$ by transformation $\tau$. Intuitively, this means that swapping the values of $y_i$ and $y_j$ does not fundamentally alter the optimization problem. Any effect of the swap can be exactly compensated by a corresponding transformation $\tau$ of the supported solutions. Thus, although a supported solution $z$ may be mapped to a different supported solution $\tau(z)$ after the parameter swap, its quality relative to the optimum remains unchanged. From the perspective of decision-making, the optimization problem therefore exhibits a symmetry with respect to parameters $y_i$ and $y_j$: interchanging them merely permutes supported solutions without changing their relative quality. We formalize this as follows.

**Proposition 2.** *Let $Z^\star$ be the set of supported solutions. Then, parameters $y_i$ and $y_j$ are homogeneous if and only if there exists a transformation $\tau : Z^\star \to Z^\star$ over solutions $z$ such that $\tau$ is an involution (i.e., $\forall z \in Z^\star : \tau(\tau(z)) = z$), and such that:*

$$\forall y \in \mathbb{R}^n \; \exists b \in \mathbb{R} \; \forall z \in Z^\star : f(z; y) = f(\tau(z); \sigma_{ij}(y)) + b$$

*Proof.* We prove the proposition by proving both directions of the equivalence separately.

($\Rightarrow$) We start by assuming that $y_i$ and $y_j$ are homogeneous, and proving that there exists a $\tau : Z^\star \to Z^\star$ such that $\forall z \in Z^\star : \tau(\tau(z)) = z$ and $\forall y \in \mathbb{R}^n \; \exists b \in \mathbb{R} \; \forall z \in Z^\star : f(z; y) + b = f(\tau(z); \sigma_{ij}(y))$.

Since $y_i$ and $y_j$ are homogeneous, the following holds:

$$\forall \hat{y}, y \in \mathbb{R}^n : Regret(\hat{y}, y) = Regret(\sigma_{ij}(\hat{y}), \sigma_{ij}(y))$$
$$\Leftrightarrow \forall \hat{y}, y \in \mathbb{R}^n : f(z^\star(\hat{y}); y) - f(z^\star(y); y) = f(z^\star(\sigma_{ij}(\hat{y})); \sigma_{ij}(y)) - f(z^\star(\sigma_{ij}(y)); \sigma_{ij}(y))$$
$$\Leftrightarrow \forall \hat{y}, y \in \mathbb{R}^n : f(z^\star(\hat{y}); y) = f(z^\star(\sigma_{ij}(\hat{y})); \sigma_{ij}(y)) + f(z^\star(y); y) - f(z^\star(\sigma_{ij}(y)); \sigma_{ij}(y)) \quad (6)$$

In the right-hand side of equation 6, the term $f(z^\star(y); y) - f(z^\star(\sigma_{ij}(y)); \sigma_{ij}(y))$ is a constant that depends on $y$. We introduce symbol $b$ to denote this constant.

$$\forall \hat{y}, y \in \mathbb{R}^n \ \exists b \in \mathbb{R} : f(z^\star(\hat{y}); y) = f(z^\star(\sigma_{ij}(\hat{y})); \sigma_{ij}(y)) + b$$

We now define transformation $\tau : Z^\star \to Z^\star$ as follows: $\tau(z^\star(y)) = z^\star(\sigma_{ij}(y))$. This transformation is only a valid function if each element of its domain is mapped onto a unique element of its codomain. In other words, there may not exist a $\hat{y}$ and $y$ for which $z^\star(\hat{y}) = z^\star(y)$ but $z^\star(\sigma_{ij}(\hat{y})) \neq z^\star(\sigma_{ij}(y))$. Since $y_i$ and $y_j$ are homogeneous, such a $\hat{y}$ and $y$ do not exist (Lemma 1), making $\tau$ a valid function. Thus:

$$\exists \tau : Z^\star \to Z^\star, \ \forall \hat{y}, y \in \mathbb{R}^n \ \exists b \in \mathbb{R} : f(z^\star(\hat{y}); y) = f(z^\star(\sigma_{ij}(\hat{y})); \sigma_{ij}(y)) + b$$

Since $Z^\star = \{z \in \Omega : \exists y \in \mathbb{R}^n : z = z^\star(y)\}$, and since $b$ depends only on $y$ (and not on $\hat{y}$):

$$\exists \tau : Z^\star \to Z^\star, \ \forall y \in \mathbb{R}^n \ \exists b \in \mathbb{R} \ \forall z \in Z^\star : f(z; y) = f(\tau(z); \sigma_{ij}(y)) + b$$

Also note that, by construction, $\tau$ is an involution, i.e., $\forall z \in Z^\star : \tau(\tau(z)) = z$. This can be shown as follows: $\forall y \in \mathbb{R}^n : \tau(\tau(z^\star(y))) = \tau(z^\star(\sigma_{ij}(y))) = z^\star(\sigma_{ij}(\sigma_{ij}(y))) = z^\star(y)$. This proves the first direction of the equivalence expressed in Proposition 2.

($\Longleftarrow$) We now prove the other direction. In other words, we assume that there exists a $\tau : Z^\star \to Z^\star$ such that $\forall z \in Z^\star : \tau(\tau(z)) = z$ and $\forall y \in \mathbb{R}^n \ \exists b \in \mathbb{R} \ \forall z \in Z^\star : f(z; y) = f(\tau(z); \sigma_{ij}(y)) + b$, and we prove that parameters $y_i$ and $y_j$ are homogeneous.

The assumption expresses that

$$\forall y \in \mathbb{R}^n \ \exists b \in \mathbb{R} \ \forall z \in Z^\star : f(z; y) = f(\tau(z); \sigma_{ij}(y)) + b$$

Since $Z^\star = \{z \in \Omega : \exists y \in \mathbb{R}^n : z = z^\star(y)\}$, it follows that:

$$\forall y, \hat{y} \in \mathbb{R}^n \ \exists b \in \mathbb{R} : f(z^\star(\hat{y}); y) = f(\tau(z^\star(\hat{y})); \sigma_{ij}(y)) + b \tag{7}$$

Next, we establish the following property:

**Claim:** $\forall y \in \mathbb{R}^n : z^\star(\sigma_{ij}(y)) = \tau(z^\star(y))$

*Proof of Claim.* We prove this by contradiction. Assume that $\exists y : z^\star(\sigma_{ij}(y)) \neq \tau(z^\star(y))$. Then, $\tau(z^\star(y))$ must be suboptimal with respect to $\sigma_{ij}(y)$:

$$\color{red} f(z^\star(\sigma_{ij}(y)); \sigma_{ij}(y)) < f(\tau(z^\star(y)); \sigma_{ij}(y))$$

By applying our primary assumption to both sides, we obtain:

$$\color{red} f(\tau(z^\star(\sigma_{ij}(y))); \sigma_{ij}(\sigma_{ij}(y))) + b < f(\tau(\tau(z^\star(y))); \sigma_{ij}(\sigma_{ij}(y))) + b$$

Because $\tau$ is an involution ($\tau(\tau(z)) = z$) and $\sigma_{ij}$ is an involution ($\sigma_{ij}(\sigma_{ij}(y)) = y$), this simplifies to:

$$\color{red} f(\tau(z^\star(\sigma_{ij}(y))); y) < f(z^\star(y); y)$$

However, this strict inequality contradicts the optimality of $z^\star(y)$ under parameter $y$. Therefore, the assumption $z^\star(\sigma_{ij}(y)) \neq \tau(z^\star(y))$ must be false, meaning $z^\star(\sigma_{ij}(y)) = \tau(z^\star(y))$.

Returning to the main proof, we can now substitute the proven claim $z^\star(\sigma_{ij}(\hat{y})) = \tau(z^\star(\hat{y}))$ back into equation 7. This gives us our starting point for establishing homogeneity:

$$\forall y, \hat{y} \in \mathbb{R}^n \ \exists b \in \mathbb{R} : f(z^\star(\hat{y}); y) = f(z^\star(\sigma_{ij}(\hat{y})); \sigma_{ij}(y)) + b$$

Next, we subtract $f(z^\star(y); y)$ from both sides of the equality:

$$\forall y, \hat{y} \in \mathbb{R}^n \ \exists b \in \mathbb{R} : f(z^\star(\hat{y}); y) - f(z^\star(y); y) = f(z^\star(\sigma_{ij}(\hat{y})); \sigma_{ij}(y)) + b - f(z^\star(y); y)$$

We then apply our main assumption to the final term $f(z^\star(y); y)$, substituting it with $f(\tau(z^\star(y)); \sigma_{ij}(y)) + b$:

$$\forall y, \hat{y} \in \mathbb{R}^n \; \exists b \in \mathbb{R} : f(z^\star(\hat{y}); y) - f(z^\star(y); y) = f(z^\star(\sigma_{ij}(\hat{y})); \sigma_{ij}(y)) + b - f(\tau(z^\star(y)); \sigma_{ij}(y)) - b$$

Using our newly proven property $\tau(z^\star(y)) = z^\star(\sigma_{ij}(y))$, we can replace the inner term. Simultaneously, $b$ and $-b$ cancel out, leaving us with:

$$\forall y, \hat{y} \in \mathbb{R}^n : f(z^\star(\hat{y}); y) - f(z^\star(y); y) = f(z^\star(\sigma_{ij}(\hat{y})); \sigma_{ij}(y)) - f(z^\star(\sigma_{ij}(y)); \sigma_{ij}(y))$$

Finally, by applying the definition of regret to both sides of the equation, we arrive at:

$$\forall y, \hat{y} \in \mathbb{R}^n : Regret(\hat{y}, y) = Regret(\sigma_{ij}(\hat{y}), \sigma_{ij}(y))$$

This proves the second direction of the equivalence, thereby proving the proposition. $\square$

Note that this proposition rules out some alternative understandings about the conditions for homogeneity that, while appearing intuitive at first, are in fact not sufficient. For instance, invariance of the optimal value under transposition (i.e., $\forall y : f(z^\star(y); y) = f(z^\star(\sigma_{ij}(y)); \sigma_{ij}(y))$) does not directly imply homogeneity of parameters $y_i$ and $y_j$. Similarly, the existence a transformation $\tau$ that preserves the optimal solution under parameter swap $\sigma_{ij}$ ($\forall y : z^\star(y) = \tau(z^\star(\sigma_{ij}(y)))$) is still insufficient to imply homogeneity. These conditions fail because homogeneity requires the conditions of Proposition 2 to apply across the entire supported solution space $Z^\star$, not merely preservation of the optimal point. We elaborate on this discussion in Appendix B, where we also provide additional examples of homogeneous parameters, and the corresponding transformations $\tau$.

The restrictive nature of these conditions reveals that parameters are truly homogeneous only under strict conditions. In other words, they are the exception rather than the rule. In practice, most realistic optimization problems instead contain heterogeneous parameters, as the strict structural symmetries required for homogeneity are rarely satisfied. In Example 1, heterogeneity was driven by non-uniform item weights. Similarly, parameter heterogeneity arises in most problems featuring varying quantities, such as blending problems with varying material restrictions, lot sizing problems with varying startup or holding costs and facility location problems with varying customer demands.

## 4.2 Degree of heterogeneity

Not all heterogeneous parameters are equally distinct. Some pairs of parameters may behave closer to homogeneous parameters than others. To capture this, we introduce a continuous metric that quantifies how asymmetrically two parameters behave under identical prediction errors. We will refer to this as the *degree* of heterogeneity. Recall from Definition 1 that for exactly homogeneous parameters, the regret is strictly invariant under the transposition $\sigma_{ij}$. We can therefore measure the degree of heterogeneity between parameter $i$ and parameter $j$ by calculating the expected violation of this invariance.

**Definition 3.** *Let $\sigma_{ij}$ be the transposition that interchanges the values of parameters $y_i$ and $y_j$, and let $\sigma^2 > 0$ characterize an error scale. The* degree of heterogeneity *at scale $\sigma$ between parameter $i$ and parameter $j$ is defined as:*

$$d_\sigma(i, j) = \mathbb{E}_{y \sim P, \epsilon \sim \mathcal{N}(0, I)} \left[ |Regret(y + \sigma\epsilon, y) - Regret(\sigma_{ij}(y + \sigma\epsilon), \sigma_{ij}(y))| \right]$$

*where the true parameters $y$ are drawn from the underlying data distribution $P$, and $\epsilon$ is a random error vector.*

This metric explicitly depends on the noise scale $\sigma$, reflecting the fact that parameter heterogeneity is tied to error magnitude. For example, an optimization problem might behave homogeneously under small mispredictions, but exhibit severe heterogeneity for larger predictive errors. Thus, in evaluating $d_\sigma(i, j)$, $\sigma$ acts as a tunable scale parameter representing the expected prediction error. In practice, $\sigma$ can simply be set to the root mean squared error of a standard prediction-focused baseline model trained to maximize predictive accuracy.

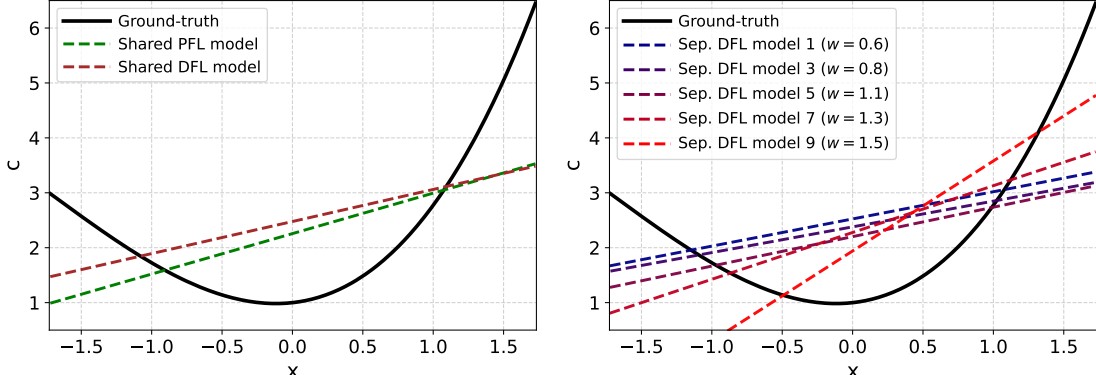

Figure 1: Learned models for a predict-then-optimize knapsack problem. Each item's value $y_i$ is unknown but correlated with a known feature $x_i$ according to a cubic relationship, which is the same for all items, and which is approximated using linear regression. Because the underlying relationship is identical for each parameter, a shared model is typically trained and used across all parameters (left panel). However, when separate parameter-specific models are trained instead (right panel), they learn a distinct mapping for each parameter, whose slopes are correlated with the associated item weights. This occurs because the parameters are heterogeneous (due to differing item weights in the constraints), and results in improved decision quality.

The expectation in $d_\sigma(i,j)$ can be empirically approximated using Monte Carlo sampling over the available training or validation data. Given a dataset of $K$ true parameter vectors $\{y^{(k)}\}_{k=1}^K$ drawn from $P$, we can sample $M$ independent noise vectors $\epsilon^{(m)} \sim \mathcal{N}(0, I)$ for each instance. The degree of heterogeneity is then computed as the sample average:

$$\hat{d}_\sigma(i,j) = \frac{1}{KM} \sum_{k=1}^K \sum_{m=1}^M \left| Regret(y^{(k)} + \sigma\epsilon^{(m)}, y^{(k)}) - Regret(\sigma_{ij}(y^{(k)} + \sigma\epsilon^{(m)}), \sigma_{ij}(y^{(k)})) \right|$$

$\hat{d}_\sigma(i,j)$ provides a diagnostic tool to evaluate the parameter heterogeneity present in a predict-then-optimize problem. When the calculated distances reveal significant parameter heterogeneity, standard shared models become less equipped to maximize downstream decision quality. In the following section, we introduce three distinct architectural adaptations and a data augmentation scheme designed to account for parameters with non-negligible heterogeneity.

## 5 Accounting for heterogeneous parameters

When a predict-then-optimize problem contains large degrees of heterogeneity, a conventional shared predictive model is not equipped to fully align its predictions with the downstream optimization problem. This is because it is forced to treat every parameter identically, and cannot distribute its prediction errors to accommodate how differently each parameter impacts downstream decision quality. Maximizing decision quality in these settings requires models capable of learning parameter-specific mappings. We illustrate this visually using the following toy problem:

**Example 2.** We generate a knapsack problem with 10 items. The total capacity of the knapsack is 5. The item weights are known, and are evenly spaced between 0.5 and 1.5. Each item value $y_i$ is unknown but correlated with a single known feature $x_i$ as follows: $y_i = 10x_i^3 - 6.5x_i + 3$ (taken from (Shah et al., 2022; 2024)). This cubic relationship is approximated using linear regression, resulting in a misspecified model that cannot fully capture the true underlying mapping. As was shown in (Elmachtoub & Grigas, 2022), DFL generally outperforms PFL under such misspecification.

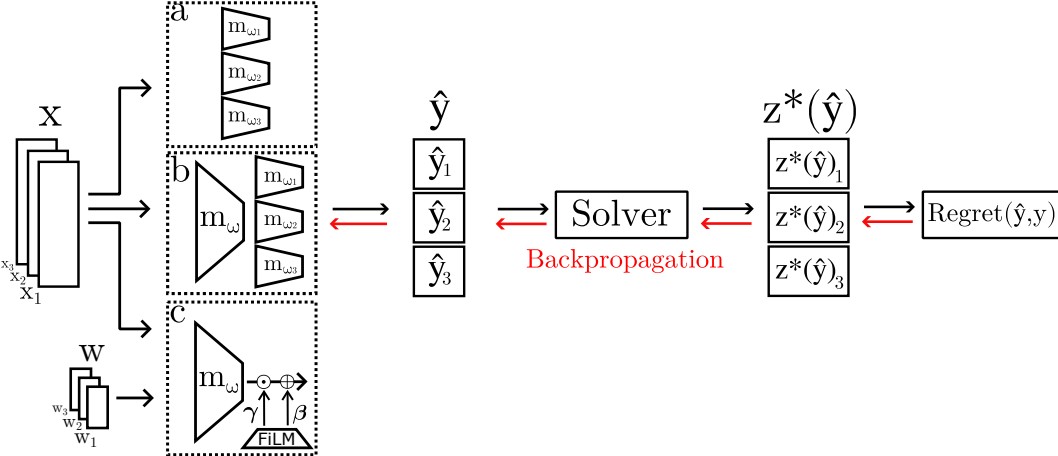

Figure 2: Three parameter-aware architectures of the predictive model: (a) a separate model per parameter, (b) a shared backbone with a separate head per parameter, and (c) a shared backbone with feature-wise linear modulation. *Comment to reviewers: one-hot parameter index encoding replace by shared model with FiLM modulation.*

> Conventionally, one would train a single model that is shared across all parameters, because each parameter depends on its feature according to the same cubic mapping. We train such a shared model with PFL and DFL. Additionally, because the parameters are heterogeneous, we use DFL to train an architecture that consists of 10 parameter-specific models.
>
> The learned models are shown in Figure 1. On the left panel, we see that the shared DFL model has learned a slightly different slope and intercept than the PFL model, resulting in marginally improved regret (8.32% vs. 8.99%). On the right panel, we show 5 of the 10 learned parameter-specific models. These models have clearly learned distinct mappings for the different parameters, with the slope of the regressor increasing as the weight of the associated item increases. This has an intuitive explanation: high-weight items should only be chosen in the knapsack solution when they have a high value, so the corresponding model mostly tries to capture the high-value region of the underlying function. This led to significantly better downstream performance, achieving a regret of 6.89%.

In what follows, we introduce three distinct architectural changes that can account for parameter heterogeneity (Definition 1), as well as a data augmentation scheme to improve data efficiency. We design these architectural alterations and the data augmentation scheme under the assumption that each parameter depends on its features according to the same true underlying mapping. This is the most common setting in prior work, and the primary setting of interest to this paper. After all, if each parameter depended on its features in a unique way, the decision to train a separate model per parameter would be the default choice, regardless of whether the parameters are heterogeneous or not.

### 5.1 Architectures

We will investigate three distinct architectural adaptations, shown in Figure 2, to learn parameter-specific mappings.

**Parameter-specific models.** A first approach is to train a separate model for each parameter, instead of one model that is shared across all parameters. In this architecture, each parameter's features are given to the associated model, i.e., $\hat{y} = [m_{\omega_1}(x_1) \ m_{\omega_2}(x_2) \ ... \ m_{\omega_n}(x_n)]$. This way, each parameter's model can learn to predict in a way that accounts for how errors on that specific parameter affect decision making. A downside, however, is that each model is only trained on a subset of the training data. If there are $d$ examples coming from the true mapping in the training set, and $n$ parameters in the optimization problem, then each model is trained on only $d/n$ examples.

**Shared backbone with parameter-specific heads.** To keep the benefit of parameter-specific models, while reducing their downside, we also consider a multi-headed architecture consisting of a shared backbone, followed by $n$ parameter-specific heads, as is common in multi-task learning (Caruana, 1997; Crawshaw, 2020; Tang & Khalil, 2022b). Each parameter prediction first passes through the shared backbone, and then through the associated head. This way, the backbone can utilize all available training data to learn the underlying mapping from features to parameter values, while each head can learn to adjust the predictions in a parameter-specific way to account for how errors impact decision quality.

**Feature-wise Linear Modulation.** Another approach to allow the model to predict in parameter-specific ways is to apply an affine transformation directly to the output of the shared model, conditioned on the known, structural attributes of each parameter (such as its associated constraint coefficients). We achieve this via a Feature-wise Linear Modulation (FiLM) mechanism (Perez et al., 2018) at the output layer. Concretely, let $w_i \in \mathbb{R}^k$ denote the vector of known structural attributes characterizing parameter $i$ (the exact nature of these attributes for our experimental benchmarks will be detailed in Section 6). An auxiliary neural network $g_\phi : \mathbb{R}^k \to \mathbb{R}^2$ processes these attributes to generate a parameter-specific scaling factor $\gamma_i$ and shifting factor $\beta_i$, such that $g_\phi(w_i) = [\gamma_i, \beta_i]^\top$. Given a base prediction produced by the shared backbone $\hat{y}_{\text{base},i} = m_\omega(x_i)$, the final modulated prediction is computed as:

$$\hat{y}_i = (\hat{y}_{\text{base},i} \cdot \gamma_i) + \beta_i \tag{8}$$

This formulation enables a single shared backbone to utilize all available training data to learn the general mapping from contextual features to parameters, while the auxiliary network calibrates the final predictions to the individual parameters. This approach provides the additional benefit that parameters with similar structural attributes, and that thus likely have a lower degree of heterogeneity, are naturally regularized to behave more similarly by the network.

### 5.2 Data augmentation

To address the reduced data efficiency inherent to parameter-specific components, we can make use of the fact that the parameters depend on their features identically, assuming that the feature–value pairs are identically distributed across parameter indices. Because of this, any value that is associated with any parameter $y_i$ in a historical instance of the optimization problem, could also have been associated with any of the other parameters. Hence we can interchange available feature-value pairs across the parameters, to create additional optimization problem instances.

We thus employ a simple offline data augmentation scheme (Algorithm 1) that can be used prior to model training. By performing a specified number of permutation rounds upfront, we construct an expanded dataset of distinct optimization problem instances. This data augmentation scheme should not be confused with ordinary data shuffling during training. While data shuffling only alters the order in which instances are presented to the model, this data augmentation scheme alters the order of parameters *within each instance*. When the parameters are heterogeneous, this transformation substantially alters the instance itself, along with its associated optimal solution.

This approach offers two distinct advantages for DFL. First, it makes up for the loss in data efficiency, because each component gets to see many more $(x_i, y_i)$ samples from the true underlying mapping. Second, by Definition 1, heterogeneous parameters are precisely those for which interchanging the values can change the regret. Consequently, permuting feature-value pairs across heterogeneous parameters alters the regret landscape of the resulting training instance and can therefore produce different gradients. In this way, the augmentation also generates additional non-redundant learning signals for DFL.

## 6 Experiments

We extensively evaluate the gains in decision quality realized by using parameter-aware architectures and data augmentation, and how they are impacted by:

---

**Algorithm 1** Offline Data Augmentation for Parameter Heterogeneity

---

**Input**: Training data $D \equiv \{(x^{(j)}, y^{(j)})\}_{j=1}^N$, Rounds of augmentation $R$
**Output**: Augmented training data $D_{\text{aug}}$

1: $D_{\text{aug}} \leftarrow D$
2: **for** $r \leftarrow 1$ **to** $R$ **do**
3:    **for** each instance $(x, y) \in D$ **do**
4:       $\pi \leftarrow$ random permutation of indices $\{1, \dots, n\}$
5:       $y_{\text{perm}} \leftarrow [y_{\pi(1)}, y_{\pi(2)}, \dots, y_{\pi(n)}]$
6:       $x_{\text{perm}} \leftarrow [x_{\pi(1)}, x_{\pi(2)}, \dots, x_{\pi(n)}]$
7:       $D_{\text{aug}} \leftarrow D_{\text{aug}} \cup \{(x_{\text{perm}}, y_{\text{perm}})\}$
8:    **end for**
9: **end for**

---

1. The amount of training data

2. The degree of parameter heterogeneity

3. The number of parameters

We investigate the impact of each of these dimensions on a different domain, in order to also evidence the general applicability of our findings. Our experiments involve both synthetic and real data. They involve an integer linear program (ILP), a mixed-integer linear program (MILP) and a continuous nonlinear optimization problem. They include linear and nonlinear predictive models. And finally, they include parameter heterogeneity originating from the constraints, the objective function, and both.

### 6.1 Experimental setup

To perform our evaluation, we train various models with differing architectures on several benchmarks (which we introduce below). The PFL models are trained to minimize the mean squared error. For continuous smooth problems, the DFL models are trained using the regret as loss function directly (Agrawal et al., 2019). For combinatorial problems, they are trained using the surrogate SPO+ loss function (Elmachtoub & Grigas, 2022). We opt for this surrogate loss because it has shown great performance across the board in comparative analyses (Tang & Khalil, 2022a; Mandi et al., 2024). However, our findings generalize to the use of other DFL loss functions, as we evidence in Appendix D. Models are trained until their regret on the validation set has not improved by at least 1% for 5 epochs. Whenever we use the data augmentation scheme, we set the number of upfront augmentation rounds to $R = 10$. All learning rates are tuned on the validation set. We report the average regret over the test set in our results. When evaluating on the test set, we always use the model weights that, throughout training, led to the best performance on an independent validation set. Reported results are always the mean and standard error of the mean, taken over 50 independent runs with different train-validation-test splits. With the exception of Table 3, we always report the average *relative* regret:

$$Rel. \; Regret(z^\star(\hat{y}), y) = \frac{f(z^\star(\hat{y}); y) - f(z^\star(y); y)}{f(z^\star(y); y)} \tag{9}$$

The relative regret is the regret equation 2 divided by true optimal value $f(z^\star(y); y)$, and expresses suboptimality as a percentage. For instance, a relative regret of 2% means that the solution produced with the predicted parameters is 2% worse than the true optimal solution. For each training method, we put the result of the best performing architecture in **bold**. Additionally, we also bold the results of architectures whose performance does not differ from the best-performing architecture with statistical significance. Further details (e.g., tuning details, architectural details, train-validation-test splits) are provided in Appendix C. All code is implemented in Python, using PyTorch (Paszke et al., 2019) and Gurobi (Gurobi Optimization, LLC, 2024). While our experimental evaluation focuses on regret, we additionally report convergence measured in training epochs and wall-clock time in Appendix E to show that the parameter-specific architectures do not introduce slower convergence. All code and data will be made available upon acceptance.

## 6.2 Benchmarks

In our evaluation, we use the following predict-then-optimize benchmarks.

### 6.2.1 Facility location over polynomial mapping

**Predict:**  The ground-truth mapping is a synthetic degree 6 polynomial that is approximated using linear regression to simulate model misspecification. This is common practice in DFL evaluations (Elmachtoub & Grigas, 2022; Mandi et al., 2022; Schutte et al., 2023; Tang & Khalil, 2022a; 2024; Mandi et al., 2024), and is especially relevant when training interpretable models (Futoma et al., 2020; Hughes et al., 2018; Sharma et al., 2021).

Concretely, the data generating process is the following. First, we generate the parameters of the true model as a vector $m \in \mathbb{R}^5$, wherein each entry is a real number uniformly sampled from $[0, 1]$. We then generate features-target pairs as follows. First, each element of feature vector $x_i \in \mathbb{R}^5$ is sampled from standard normal distribution $\mathcal{N}(0, 1)$. Then, the corresponding target value $y_i \in \mathbb{R}$ is generated as $y_i = \frac{1}{3.5^6} \left( 1 + (\frac{m^\top x}{5} + 3)^6 \right)$.

**Optimize:**  Given a set of potential facility locations $\mathcal{I}$ and a set of customers $\mathcal{J}$, the capacitated facility location problem asks to minimize the total operation cost while satisfying the customer demands of a single product. Let $d_j$ be the demand of customer $j$, $f_i$ be the fixed cost of opening a facility at location $i$, $y_{ij}$ be the transportation cost for one unit of product from a facility at location $i$ to customer $j$, and $u_i$ be the capacity of a facility located at $i$. Also, let $z_i$ be a binary decision variable that denotes whether a facility is opened at location $i$, and let $z_{ij}$ be a continuous decision variable denoting the fraction of demand of customer $j \in \mathcal{J}$ served from a facility located at $i \in \mathcal{I}$. Then, the capacitated facility location problem can be formalized as follows:

$$\min \quad \sum_{i \in \mathcal{I}} \sum_{j \in \mathcal{J}} y_{ij} d_j z_{ij} + \sum_{i \in \mathcal{I}} f_i z_i \tag{10}$$

$$\text{s.t.} \quad \sum_{i \in \mathcal{I}} z_{ij} = 1 \text{ for all } j \in \mathcal{J} \tag{11}$$

$$\sum_{j \in \mathcal{J}} d_j z_{ij} \leq u_i z_i \text{ for all } i \in \mathcal{I} \tag{12}$$

$$z_i \in \{0, 1\}, z_{ij} \geq 0 \text{ for all } i \in \mathcal{I} \text{ and } j \in \mathcal{J}. \tag{13}$$

The objective (10) minimizes the total cost of transportation and the fixed costs for the opened facilities. Constraint (11) ensures that the demand of each customer is satisfied. Constraint (12) guarantees that facility capacities are respected and that only the opened facilities are used. Constraint (13) is the domain constraint for the decision variables.

In our experiments, we use 5 facilities and 10 customers. The transportation costs $y_{ij}$ are unknown and to be predicted by the predictive model. The other quantities in the optimization problem are known, and generated as follows. The capacities $u_i$ are evenly spaced between 5 and 15. Each demand $d_j$ is set to $max(0, D)$, where $D$ is sampled (independently for each customer's demand) from $\mathcal{N}(2.5, 1.25)$, which is the normal distribution with mean 2.5 and standard deviation 1.25. Each setup cost $f_i$ is sampled from $U[3, 10]$, where $U$ denotes the uniform distribution. When using the FiLM architecture as predictive model, the conditioning network receives the known structural attributes associated with each parameter, which consist of the facility capacity $u_i$, the customer demand $d_j$, and the fixed facility setup cost $f_i$.

### 6.2.2 Knapsack over California housing prices

**Predict:**  This prediction task is taken from a common machine learning benchmark (Pace & Barry, 1997), which is offered as a benchmark in the scikit-learn Python package (Pedregosa et al., 2011). In this benchmark, the median house prices of districts in California must be predicted from 8 correlated local features, including spatial features, features about the districts' populations and other aggregate housing statistics. We use linear predictive model to predict the median house prices from these features. To construct predict-then-optimize instances, we first randomly shuffle the districts in the dataset and partition them into disjoint

training, validation, and test sets. We then generate optimization instances by randomly sampling 40 districts from the corresponding split, with each district representing a single item in the knapsack problem. All training sets contained 250 optimization instances (i.e., 10,000 feature-parameter pairs in total), while the validation and test sets each contained 129 optimization instances (i.e., 5,160 feature-parameter pairs).

**Optimize:** Given that there is a set of items $\mathcal{I} = \{1, \dots, n\}$ where item $i \in \mathcal{I}$ has a weight $w_i$ and value $y_i$, the knapsack problem asks to find the subset of all items with maximum total value without exceeding the knapsack capacity $W$. Let $z_i$ be a binary decision variable that denotes whether item $i$ is selected. The knapsack problem is then formalized as follows:

$$\max \quad \sum_{i \in \mathcal{I}} y_i z_i \tag{14}$$

$$\text{s.t.} \quad \sum_{i \in \mathcal{I}} w_i z_i \leq W \tag{15}$$

$$z_i \in \{0, 1\} \text{ for all } i \in \mathcal{I}. \tag{16}$$

The objective function (14) maximizes the total value of selected items. Constraint (15) ensures that the knapsack capacity is not exceeded. Constraint (16) is the domain constraint for the decision variables.

In our experiments, the problem contains 40 items. The total capacity is 50. The item values $y_i$ are unknown and to be predicted by the predictive model. The item weights are evenly spaced over a range which we vary to control the degree of heterogeneity. In the homogeneous parameters setting, each weight is set to 2.5. In the heterogeneous parameters setting with a small range of weights, the weights are evenly spaced between 1.25 and 3.75. In the heterogeneous parameters setting with a large range of weights, the weights are evenly spaced between 0 and 5. The structural attribute provided to the FiLM model is the known item weight $w_i$.

### 6.2.3 Multi-item newsvendor over Friedman data

**Predict:** The synthetic Friedman #1 regression dataset was first described in (Friedman, 1991). A generator is provided in scikit-learn (Pedregosa et al., 2011). Each features-target pair is generated as follows. First, 5 features are sampled uniformly from the range $[0, 1]$. Then, the target value $y_i$ follows from its features through the nonlinear mapping $y_i = 10 \sin(\pi x_1 x_2) + 20(x_3 - 0.5)^2 + 10x_3 + 5x_4 + 2\epsilon$, where $\epsilon$ is sampled from $N(0, 1)$

**Optimize:** There is a set of items $\mathcal{I} = \{1, \dots, n\}$. The problem involves deciding how much of each item to produce. There is an overall budget, limiting the total production. Overproduction and underproduction have distinct costs in the objective, and the ratio between these costs differs from item to item. Let $z_i$ be the produced quantity of item $i$, let $y_i$ be the demand of item $i$, and let $u_i$ and $o_i$ be the costs of underproducing and overproducing item $i$ (with respect to demand $y_i$), respectively. Then, the multi-item newsvendor problem is formalized as follows:

$$\min \quad \sum_{i \in \mathcal{I}} \left[ u_i \max(y_i - z_i, 0) + o_i \max(z_i - y_i, 0) \right] \tag{17}$$

$$\text{s.t.} \quad \sum_{i \in \mathcal{I}} z_i \leq W \tag{18}$$

$$z_i \geq 0 \text{ for all } i \in \mathcal{I}. \tag{19}$$

The objective function (17) minimizes the total cost of underproduction and overproduction of the various items. Constraint (18) ensures that the total production capacity is not exceeded. Constraint (19) ensures that production cannot be negative.

In our experiments, we used the multi-item newsvendor problem to investigate the effect of the number of parameters in the optimization problem. These parameters represented the item demands. We varied the number of parameters, and thus items, by considering problems with 10, 50 and 100 items. The total production capacity was set such that, on average, about 75% of the demand in a problem instance could be

Table 1: The effect of the training set size, evaluated on a facility location problem using a polynomial mapping

| Method | Architecture | Relative regret (%) | | | |
|---|---|---|---|---|---|
| | | 10 instances | 20 instances | 100 instances | 200 instances |
| PFL | Shared Model | $\mathbf{3.98 \pm 0.34}$ | $\mathbf{3.95 \pm 0.32}$ | $\mathbf{3.90 \pm 0.29}$ | $\mathbf{3.85 \pm 0.30}$ |
| | Separate Models | $15.57 \pm 0.84$ | $11.47 \pm 0.63$ | $5.97 \pm 0.40$ | $4.48 \pm 0.33$ |
| | Separate Heads | $13.27 \pm 2.42$ | $11.18 \pm 2.04$ | $5.55 \pm 0.42$ | $4.48 \pm 0.36$ |
| | Shared + FiLM | $17.71 \pm 2.99$ | $10.86 \pm 2.01$ | $4.56 \pm 0.38$ | $4.16 \pm 0.33$ |
| DFL | Shared Model | $\mathbf{2.95 \pm 0.22}$ | $\mathbf{2.80 \pm 0.21}$ | $2.71 \pm 0.18$ | $2.67 \pm 0.18$ |
| | Separate Models | $27.46 \pm 1.03$ | $13.64 \pm 0.44$ | $2.97 \pm 0.14$ | $2.18 \pm 0.12$ |
| | Separate Heads | $15.67 \pm 1.27$ | $5.68 \pm 0.26$ | $2.25 \pm 0.14$ | $1.94 \pm 0.13$ |
| | Shared + FiLM | $6.68 \pm 1.54$ | $\mathbf{4.08 \pm 1.25}$ | $\mathbf{1.74 \pm 0.12}$ | $\mathbf{1.68 \pm 0.12}$ |
| DFL | Shared Model | $2.65 \pm 0.17$ | $2.60 \pm 0.16$ | $2.83 \pm 0.19$ | $2.84 \pm 0.19$ |
| with data | Separate Models | $2.97 \pm 0.16$ | $2.20 \pm 0.12$ | $1.73 \pm 0.11$ | $1.71 \pm 0.10$ |
| augmentation | Separate Heads | $2.23 \pm 0.14$ | $1.92 \pm 0.13$ | $1.71 \pm 0.11$ | $1.67 \pm 0.11$ |
| | Shared + FiLM | $\mathbf{1.73 \pm 0.12}$ | $\mathbf{1.62 \pm 0.11}$ | $\mathbf{1.56 \pm 0.11}$ | $\mathbf{1.52 \pm 0.10}$ |

met. The underproduction unit costs $u_i$ were evenly spaced between 1 and 10, while the overproduction unit costs $o_i$ were evenly spaced over the same range, but in reverse order, i.e., they were evenly spaced between 10 and 1. The conditioning network for FiLM is provided with the structural cost parameters of each item, namely the underproduction unit cost $u_i$ and the overproduction unit cost $o_i$.

## 6.3 Results and discussion

**Amount of training data.** We investigate how our the architectures with parameter-specific mappings compare to the conventional use of a shared model, in function of the amount of training data available. We do so using the facility location problem over a polynomial mapping, as the synthetic nature of this mapping allows us to vary the amount of available training data arbitrarily. As is conventional when using this mapping in DFL evaluations, the predictive models are fully linear, introducing model misspecification. The results can be found in Table 1.

The first four rows of this table (as well as of later tables) show that our proposed architectures do not have a significant positive impact when the model is trained to maximize accuracy using PFL. This is expected, since each parameter depends on features according to the same true underlying mapping, meaning that learning parameter-specific mappings does not benefit predictive accuracy.

The next four rows show that better decision quality (i.e., lower regret) can be achieved by training with DFL instead of PFL. More remarkably, they show that, when sufficient training data is available, all architectures that incorporate parameter-specific components outperform the conventional shared model, despite the fact that all parameters are generated by the same underlying mapping from features to coefficients. Among these architectures, the shared backbone with FiLM modulation consistently achieves the lowest regret when enough data is available. Thus, we observe that, although parameter-specific components offer no advantage from a predictive perspective, DFL is able to exploit it to improve decision quality. This suggests that the benefits of parameter-specific components arise not from better approximating the true mapping, but from learning parameter-specific prediction errors that are advantageous for the downstream optimization problem.

The final four rows show that further improvements can be realized by the use of our data augmentation scheme, in which the parameter vectors available in the training data are permuted in random ways. As expected, this is especially important in the low data regime, although it also leads to improvements when more data is available.

**Degree of parameter heterogeneity.** We investigate the difference in performance between the architectures for problems with varying degrees of parameter heterogeneity, using knapsack problems over housing districts in California. To vary the degree of heterogeneity, we use knapsack problems with varying weight distributions. To evaluate the degrees of heterogeneity of these problems, we must first set an error scale (Definition 3). To pick an appropriate scale that reflects the expected error magnitude, we compute the RMSE of a baseline shared model trained to maximize accuracy, and set the error scale $\sigma$ to this value. This led to a noise standard deviation of $\sigma = 2.06$.

We then construct a sequence of knapsack problem families with increasing levels of weight variability. The first family is fully homogeneous, where all item weights are fixed at 2.5, resulting in an average degree of parameter heterogeneity of $d_\sigma = 0.000$ across all parameter pairs. To introduce moderate heterogeneity, we generate a second family with weights evenly spaced between $[1.25, 3.75]$, which leads to an average heterogeneity of $d_\sigma = 0.696$. Finally, we consider a high-heterogeneity setting with weights spaced between $[0, 5]$, leading to an average degree of $d_\sigma = 0.849$. The regrets on these different problems can be found in Table 2.

Again, the first three rows PFL results evidence the fact that each parameter depends on its features in the same way. The middle rows show the results for models trained with DFL. DFL improves the regret across all architectures and heterogeneity levels. When the parameters are homogeneous, the architectures with parameter-specific components do not outperform the conventional shared model, which is to be expected. However, when there is some heterogeneity in the parameters, the shared backbone with separate parameter-specific heads perform best, while separate models and a shared backbone with FiLM modulation also outperform the fully shared model, although the difference is more modest than in the previous domain. Increasing the heterogeneity further increases this gap in performance. These results are in line with our expectation: it is the heterogeneity of the parameters that motivates the introduction of parameters-specific components in the model.

On this domain, data augmentation has only a modest and inconsistent effect on performance, indicating that there is already sufficient data available.

The simplicity of the knapsack problem also allows us to get some further insight into what these components have specifically learned. To this end, we inspect the way the prediction errors are distributed for the different items, by computing the Pearson correlation coefficient between the prediction errors (on the test set) and the weights (i.e., constraint coefficients) of the items for which the predictions are respectively made. For the models trained with DFL in the slightly heterogeneous setting, this correlation coefficient is 0.001 for the shared architecture, $-0.402$ for the separate models, and $-0.220$ for the shared backbone with separate heads. The negative correlation coefficients for our proposed architectures mean that these models have learned to, on average, *underestimate* more the relative value of high-weight items, and *overestimate* more the relative value of low-weight items. This makes sense: high-weight items are more costly, and thus chosen less often on average. Using our proposed architectures, the models have learned to distribute their prediction errors in parameter-specific ways.

**Number of parameters.** We investigate how the number of parameters in the optimization problem affects performance. We do so using a multi-item newsvendor problem over the Friedman #1 dataset. When changing the number of parameters, we fix the total number of samples from the true underlying mapping that are available in the training set, so as to isolate the specific effect of the number of parameters. In other words, when we increase the number of parameters, this does not alter the amount of training data the model has access to. Also note that the true optimal values are quite small in this domain, making the denominator in the *relative* regret computation close to zero. For this reason, we instead directly report the regret equation 2 on this benchmark. The predictive models are neural networks with a single hidden layer of 32 neurons, utilizing ReLU activation. The results are shown in Table 3.

For the smallest number of parameters, the results are in line with what was discussed before: training with PFL does not benefit from learning parameter-specific mappings, training with DFL outperforms PFL, and training with DFL *does* benefit from parameter-specific mappings. Data augmentation improves performance further.

Table 2: The effect of homogeneous vs. heterogeneous parameters, evaluated on a knapsack problem using housing data (50 seeds)

| Method | Architecture | Relative regret (%) | | |
|---|---|---|---|---|
| | | **Homogeneous** | **Heterogeneous** | |
| | | | **Small range** | **Large range** |
| PFL | Shared Model | $\mathbf{5.36 \pm 0.05}$ | $\mathbf{4.49 \pm 0.04}$ | $\mathbf{3.31 \pm 0.03}$ |
| | Separate Models | $5.50 \pm 0.04$ | $4.79 \pm 0.03$ | $3.62 \pm 0.03$ |
| | Separate Heads | $5.72 \pm 0.06$ | $4.79 \pm 0.04$ | $3.55 \pm 0.03$ |
| | Shared + FiLM | $\mathbf{5.36 \pm 0.05}$ | $\mathbf{4.50 \pm 0.04}$ | $3.36 \pm 0.03$ |
| DFL | Shared Model | $\mathbf{4.43 \pm 0.04}$ | $4.04 \pm 0.04$ | $3.13 \pm 0.03$ |
| | Separate Models | $4.69 \pm 0.03$ | $3.97 \pm 0.03$ | $2.97 \pm 0.03$ |
| | Separate Heads | $\mathbf{4.46 \pm 0.03}$ | $\mathbf{3.90 \pm 0.03}$ | $\mathbf{2.89 \pm 0.02}$ |
| | Shared + FiLM | $\mathbf{4.42 \pm 0.04}$ | $3.98 \pm 0.04$ | $2.96 \pm 0.03$ |
| DFL | Shared Model | $\mathbf{4.38 \pm 0.04}$ | $4.09 \pm 0.04$ | $3.19 \pm 0.03$ |
| with data | Separate Models | $4.62 \pm 0.03$ | $3.89 \pm 0.03$ | $2.78 \pm 0.03$ |
| augmentation | Separate Heads | $\mathbf{4.41 \pm 0.04}$ | $\mathbf{3.78 \pm 0.03}$ | $\mathbf{2.71 \pm 0.03}$ |
| | Shared + FiLM | $\mathbf{4.37 \pm 0.04}$ | $4.02 \pm 0.04$ | $3.00 \pm 0.03$ |

Table 3: The effect of output dimension size, evaluated on a newsvendor problem using absolute regret

| Method | Architecture | Regret | | |
|---|---|---|---|---|
| | | **10 parameters** | **30 parameters** | **50 parameters** |
| PFL | Shared Model | $\mathbf{69.95 \pm 2.59}$ | $\mathbf{218.30 \pm 8.87}$ | $\mathbf{410.08 \pm 18.82}$ |
| | Separate Models | $78.01 \pm 1.87$ | $290.10 \pm 6.78$ | $531.71 \pm 7.39$ |
| | Separate Heads | $\mathbf{71.27 \pm 1.87}$ | $\mathbf{230.90 \pm 5.72}$ | $\mathbf{505.89 \pm 21.34}$ |
| | Shared + FiLM | $\mathbf{66.44 \pm 0.54}$ | $\mathbf{209.91 \pm 4.16}$ | $\mathbf{402.73 \pm 21.67}$ |
| DFL | Shared Model | $\mathbf{70.67 \pm 2.15}$ | $218.54 \pm 8.67$ | $\mathbf{347.09 \pm 14.08}$ |
| | Separate Models | $88.37 \pm 3.64$ | $340.50 \pm 20.26$ | $894.79 \pm 37.49$ |
| | Separate Heads | $\mathbf{83.92 \pm 10.41}$ | $259.78 \pm 13.28$ | $580.91 \pm 65.75$ |
| | Shared + FiLM | $\mathbf{61.61 \pm 2.10}$ | $\mathbf{176.65 \pm 2.69}$ | $\mathbf{310.71 \pm 14.02}$ |
| DFL | Shared Model | $66.60 \pm 0.47$ | $195.04 \pm 0.85$ | $339.74 \pm 7.09$ |
| with data | Separate Models | $98.71 \pm 7.48$ | $266.20 \pm 14.25$ | $513.34 \pm 49.25$ |
| augmentation | Separate Heads | $64.07 \pm 1.27$ | $215.78 \pm 9.56$ | $424.18 \pm 27.35$ |
| | Shared + FiLM | $\mathbf{57.69 \pm 0.46}$ | $\mathbf{172.76 \pm 1.84}$ | $\mathbf{285.86 \pm 1.99}$ |

When the number of parameters increases, the importance of data augmentation grows. This is because increasing the number of parameters also shrinks the segments of the dataset that are used to train the parameter-specific mappings with. The use of data augmentation makes up for this.

In this domain, neither the separate models nor separate heads consistently outperform the shared model. In fact, their performance degrades substantially as the dimensionality grows. The shared backbone with FiLM modulation is the only architecture that consistently improves upon the shared model, with the advantage becoming more pronounced for larger numbers of parameters. We attribute this to the higher expressivity of the predictive models used in this benchmark compared to the previous experiments. When the models are more complex, training separate models or heads can become more difficult and data inefficient. FiLM, on the other hand, uses a single shared auxiliary network for parameter conditioning, while using a shared backbone for prediction. As a result, increasing the number of parameters in the optimization problem does not increase the number of learnable parameters in the neural network.

# 7 Conclusion and future work

In this paper, we investigated an understudied aspect of DFL: that mispredictions can affect downstream decision quality in different ways for different parameters. We formalized this by distinguishing between homogeneous and heterogeneous parameters, and provided a theoretical result that characterizes exactly when parameters are heterogeneous. This characterization shows that homogeneity requires a strong symmetry property of the optimization problem: there must exist an involutive transformation on supported solutions under which the objective is invariant, up to an additive constant, when parameters are permuted. As a consequence, homogeneous parameters arise only under restrictive structural conditions, whereas heterogeneous parameters are the norm in realistic optimization problems that involve non-uniform constraints, costs, demands, or capacities. We then introduced a continuous metric to quantify the degree of parameter heterogeneity in practical settings.

We then showed that accounting for heterogeneous parameters in the architecture of the predictive model can significantly improve decision quality. To this end, we investigated three architectural adaptations, and an offline data augmentation scheme to improve data efficiency. Our experiments confirm that, in the presence of parameter heterogeneity, parameter-aware architectures outperform a conventional shared model. The results also showed that – under parameter heterogeneity – data augmentation leads to further improvements, especially when little data is available, or when the problem contains many parameters.

A promising direction for future work is adaptive parameter clustering. Using the degree of heterogeneity metric ($d_\sigma$), models could group similar parameters and share weights among them, balancing data efficiency against parameter-specific expressiveness. Another promising direction is the extension to a setting in which the parametric optimization problem itself varies across instances, for example when constraint coefficients or feasible regions are instance-dependent. Finally, given the data limitations that often arise in DFL applications, a broader exploration of data augmentation techniques may be valuable. For example, when only a few samples from the true mapping are available, different combinations of these samples within the optimization problem may provide additional signal to the predictive model.

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

## A  Proof of lemma

In this section, we prove the supporting lemma. For clarity, we assume that $z^\star(y)$ is uniquely defined for each $y$, though the proofs can be readily extended to cases with multiple optimal solutions.

**Lemma 3.** *Let $y_i$ and $y_j$ be two homogeneous parameters. Then, there do not exist any two parameter vectors $y$ and $y'$ for which $z^\star(y) = z^\star(y')$ but $z^\star(\sigma_{ij}(y)) \neq z^\star(\sigma_{ij}(y'))$.*

*Proof.* (By contradiction) Assume that parameters $y_i$ and $y_j$ are homogeneous, and that there *do* exist two parameter vectors $y$ and $y'$ for which $z^\star(y) = z^\star(y')$ but $z^\star(\sigma_{ij}(y)) \neq z^\star(\sigma_{ij}(y'))$

Then, since $z^\star(\sigma_{ij}(y)) \neq z^\star(\sigma_{ij}(y'))$, it must hold that

$$f(z^\star(\sigma_{ij}(y')); \sigma_{ij}(y')) < f(z^\star(\sigma_{ij}(y)); \sigma_{ij}(y')) \tag{20}$$

Also, since $y_i$ and $y_j$ are homogeneous, the following holds:

$$Regret(y, y') = Regret(\sigma_{ij}(y), \sigma_{ij}(y')) \tag{21}$$

$$\Rightarrow f(z^\star(y); y') - f(z^\star(y'); y') = f(z^\star(\sigma_{ij}(y)); \sigma_{ij}(y')) - f(z^\star(\sigma_{ij}(y')); \sigma_{ij}(y')) \tag{22}$$

$$\Rightarrow f(z^\star(y'); y') - f(z^\star(y'); y') = f(z^\star(\sigma_{ij}(y)); \sigma_{ij}(y')) - f(z^\star(\sigma_{ij}(y')); \sigma_{ij}(y')) \tag{23}$$

$$\Rightarrow f(z^\star(\sigma_{ij}(y)); \sigma_{ij}(y')) = f(z^\star(\sigma_{ij}(y')); \sigma_{ij}(y')) \tag{24}$$

Equation 21 follows from the definition of homogeneous parameters. Equation 22 follows from the definition of regret. Equation 23 follows from the assumption that $z^\star(y) = z^\star(y')$. Finally, equation 24 is derived by rearranging terms.

Equation 24 is in contradiction with inequality 20. This contradiction arises from our initial assumption when $y_i$ and $y_j$ are homogeneous, there can exist parameter vectors $y$ and $y'$ such that $z^\star(y) = z^\star(y')$ but $z^\star(\sigma_{ij}(y)) \neq z^\star(\sigma_{ij}(y'))$. Therefore, the assumption is false, and the lemma is proved. $\qquad\square$

## B    Additional discussion of theoretical results

We start this discussion by illustrating how Proposition 2 applies to Example 1. We repeat the proposition here:

**Proposition 4.** *Let $Z^\star = \{z \in \Omega : \exists y \in \mathbb{R}^n : z = z^\star(y)\}$ be the set of* supported *solutions. Then, parameters $y_i$ and $y_j$ are homogeneous if and only if there exists a transformation $\tau : Z^\star \to Z^\star$ over solutions $z$ such that $\tau$ is an involution (i.e., $\forall z \in Z^\star : \tau(\tau(z)) = z$), and such that:*

$$\forall y \in \mathbb{R}^n \; \exists b \in \mathbb{R} \; \forall z \in Z^\star : f(z; y) = f(\tau(z); \sigma_{ij}(y)) + b$$

Now consider again the parameteric optimization problem from Example 1:

$$
\begin{aligned}
z^\star(y) = \arg\max \quad & y_1 z_1 + y_2 z_2 + y_3 z_3 \\
\text{s.t.} \quad & z_1 + z_2 + 2z_3 \leq 2 \\
& z_1, z_2, z_3 \in \{0, 1\}
\end{aligned}
$$

In this problem, parameters $y_1$ and $y_2$ are homogeneous. Here, $\tau$ is simply the transposition that interchanges the values of $z_1$ and $z_2$, while $b = 0$ for all $y$. Then, for example, for $y = [1\ 2\ 3]$, and $z = [0\ 1\ 0]$: $f(z; y) = [0\ 1\ 0][1\ 2\ 3]^\top = 2$, while $f(\tau(z); \sigma_{ij}(y)) + b = [1\ 0\ 0][2\ 1\ 3]^\top + 0 = 2$.

In Example 1, the homogeneous parameters were associated with items of identical weight, while heterogeneous parameters were associated with items of differing weight. However, the distinction is not always as easily observable. It is possible to construct pathological cases in which the parameters appear heterogeneous but are, in fact, homogeneous. Take, for example, the following problem:

$$
\begin{aligned}
z^\star(y) = \arg\max \quad & y_1 z_1 + y_2 z_2 + y_3 z_3 \\
\text{s.t.} \quad & 3z_1 + 2z_2 + z_3 \leq 1 \\
& z_1, z_2, z_3 \in \{0, 1\}
\end{aligned}
$$

Here, $y_1$ and $y_2$ are homogeneous (with the same $\tau$ and choice of $b$ as before), despite having differing weights in the constraint. This intuitively makes sense: since $z_1$ and $z_2$ both must be 0, it does not make a difference to the regret whether a certain prediction error is made on $y_1$ or on $y_2$.

While in the previous two examples, additive term $b$ was always 0; this term may not be left out of the proposition. We show this by means of the following example:

$$
\begin{aligned}
z^\star(y) = \arg\max \quad & y_1 z_1 + y_2 z_2 \\
\text{s.t.} \quad & z_1 = 1 \\
& z_2 = 0
\end{aligned}
$$

Parameters $y_1$ and $y_2$ are clearly homogeneous, since the only feasible solution is $z = [1\ 0]$, and thus the regret is always 0. However, to apply the proposition, $\tau$ is the identity function, while $\forall y : b = y_1 - y_2$. For instance, take $y = [3\ 1]$, for which $b = 2$. Then, $f(z; y) = 3$, while $f(\tau(z); \sigma_{12}(y)) + b = 1 + 2 = 3$

Finally, since the proposition captures the necessary and sufficient conditions for homogeneity to occur, it also rules out several competing explanations for homogeneity. For instance, when the optimal value of the problem is invariant to transposition $\sigma_{ij}$ (i.e., $\forall y : f(z^\star(y) = f(z^\star(\sigma_{ij}(y))))$), this does not directly imply homogeneity of parameters $y_i$ and $y_j$. Similarly, even if there exists a transformation $\tau$ that preserves the optimal solution with respect to $\sigma_{ij}$ (i.e., $\forall y : z^\star(y) = \tau(z^\star(\sigma_{ij}(y))))$), this still does not imply homogeneity. We illustrate both using the following unconstrained optimization problem:

$$z^\star(y) = \arg\min \quad |y_1 - z_1| + 2|y_2 - z_2|$$

In this problem, both $\forall y : f(z^\star(y); y) = f(z^\star(\sigma_{ij}(y)); \sigma_{ij})$ and $\exists \tau : \forall y : z^\star(y) = \tau(z^\star(\sigma_{ij}(y)))$ hold. The former holds because for every $y$, the optimal value is 0. The latter holds for $\tau$ equal to the transposition that interchanges the values of $z_1$ and $z_2$, because for any $y$, the optimal solution $z^\star(y) = [y_1\ y_2]$. Thus, after applying $\sigma_{12}$, the optimal solution becomes $\tau(z^\star(y)) = [y_2\ y_1]$.

Still, parameters $y_1$ and $y_2$ are heterogeneous. Take, for example, $y = [1\ 1]$ and $\hat{y} = [2\ 3]$. Now, $Regret(\hat{y}, y) = 5$, while $Regret(\sigma_{12}(y), \sigma_{12}(\hat{y})) = 4$.

## C  Detailed experimental setup

In this section, we provide additional details about our experimental setup. We begin with general considerations that apply to all three experiments before discussing specifics for each one.

To perform our evaluation, we trained various models with differing architectures on several benchmarks. All models were trained using the Adam (Kingma & Ba, 2014) optimizer. The learning rates were tuned separately for each method-architecture-benchmark combination, where validation set regret was used to decide which learning rate performed best. Considered values were $0.001, 0.01, 0.1$ and $1$. The best-performing learning rate turned out to be $0.01$ in each experiment. Models are trained until their regret on the validation set has not improved by at least 1% for 5 epochs. Whenever we use the data augmentation scheme, we set the number of upfront augmentation rounds to $R = 10$. The batch size was chosen to be 16. Note that this refers to the number of optimization problem instances. Thus, for an optimization problem containing 10 variables, a batch size of 16 means that 160 samples from the ground-truth mapping from features to parameters are involved in one weight update.

Reported results are always the mean test set regret, and the standard error of the mean, taken over 50 independent runs with different train-validation-test splits. When evaluating on the test set, we always use the model that, throughout training, performed best on the validation set. In the tables of results, for each training method, we put the result of the best performing architecture in **bold**. Additionally, we also bold the results of architectures whose performance does not differ significantly from the best-performing architecture. To assess whether an architecture performs significantly worse than the best architecture, we conduct paired t-tests on the test regrets across the 50 dataset seeds. The base significance level is set to $\alpha = 0.05$, to which we apply the Holm-Bonferroni correction to account for multiple comparisons.

### C.1  Experiment 1: Effect training set size

This experiment was performed using the polynomial mapping as ground-truth features-parameters mapping, and using the facility location problem as optimization problem. Each instance of the optimization problem contained 5 facilities and 10 customers, meaning that 50 samples from the ground-truth mapping were used per instance. The training set sizes were varied in the experiment, and are reported in Table 1. The validation set always contained 250 instances of the optimization problem, while the test set always contained 1000 instances. The models were linear models, containing one hidden layer. Although the hidden layer does not add to the representational capacity of the model, it allows for the introduction of variable-specific heads. The hidden layer contained 5 hidden units. In other words, the first layer was a $\mathbb{R}^5 \to \mathbb{R}^5$ mapping

and the second layer was a $\mathbb{R}^5 \to \mathbb{R}$ mapping. For the shared architecture, both layers are shared for all parameters. For the separate models architecture, each parameter has its own dedicated two-layer model. For the architecture consisting of a shared backbone and separate heads, the first layer is shared for all parameters, while each parameter has its own dedicated second layer.

### C.2    Experiment 2: Effect homogeneous vs. heterogeneous parameters

This experiment was performed using the California housing prices mapping as ground-truth features-parameters mapping, and using the knapsack problem as optimization problem. Each instance of the knapsack problem contained 40 items. All training sets contained 250 instances of the optimization problem (i.e., $40 \times 250 = 10000$ features-parameter pairs). All validation and test sets contained 129 instances of the optimization problem. The models were linear models, this time with a hidden layer size of 8, to match the 8 input features.

### C.3    Experiment 3: Effect number of variables

This experiment was performed using the Friedman #1 mapping as ground-truth features-parameters mapping, and using the multi-item newsvendor problem as optimization problem. The number of parameters in the optimization problem was varied in the experiment between 10, 50 and 100. Because of the varying number of decision variables, the total number of optimization problem instances in the training set also had to be varied, in order to keep the total number of samples from the ground-truth mapping in the training set constant. For 10 parameters, the training set contained 500 instances. For 50 parameters, it contained 100 instances. For 100 parameters, it contained 50 instances. The remaining instances out of a total of 1000 were divided equally between the validation and test set. The models were nonlinear models with one hidden layer containing 32 hidden units and ReLU activation. The architecture with a shared backbone, the first $5 \to 32$ layer was shared, and the subsequent $32 \to 1$ layers were parameter-specific.

### C.4    FiLM architecture

When using the FiLM architecture, the structural attributes vector $w_i$ is normalized before being passed to the auxiliary network $g_\phi$.

- In the facility location problem, $w_i \in \mathbb{R}^3$ concatenates the capacity, demand, and setup cost associated with that specific parameter coordinate.

- In the knapsack problem, $w_i \in \mathbb{R}^1$ is the scalar item weight.

- In the newsvendor problem, $w_i \in \mathbb{R}^2$ contains the underproduction and overproduction unit costs.

The auxiliary network $g_\phi$ is implemented as a small two-layer multi-layer perceptron with a hidden size of 16 and ReLU activations, outputting the parameter-specific scaling factor $\gamma_i$ and shifting factor $\beta_i$.

## D    Additional results for other DFL losses

To strengthen the evidence for the generality of our conclusions, we provide some additional results including other DFL losses below. We use the parametric knapsack problem in combination with a ground-truth polynomial mapping of degree 6. The knapsack weights are evenly spaced between 0 and 2, leading to parameter heterogeneity in the objective. We average the results over 10 seeds. Additional included DFL methods are the PFYL method (Berthet et al., 2020) and the Identity method (Sahoo et al., 2022).

## E    Convergence speed statistics

In this section, we present the convergence speed and computational runtimes for the empirical evaluations discussed in Section 6. To evidence the efficiency of the parameter-aware architectures, we record performance

Table 4: Results for some additional DFL losses

| Loss | Architecture | Relative regret (%) |
|------|-------------|---------------------|
| MSE | Shared Model | **0.26 ± 0.01** |
| | Separate Models | 0.37 ± 0.01 |
| | Separate Heads | 0.32 ± 0.01 |
| SPO+ | Shared Model | 0.25 ± 0.01 |
| | Separate Models | 0.10 ± 0.01 |
| | Separate Heads | **0.03 ± 0.01** |
| PFYL | Shared Model | 0.30 ± 0.01 |
| | Separate Models | 0.12 ± 0.01 |
| | Separate Heads | **0.04 ± 0.01** |
| Identity | Shared Model | 0.36 ± 0.03 |
| | Separate Models | 0.17 ± 0.01 |
| | Separate Heads | **0.09 ± 0.01** |

across 50 independent runs under a validation-based early stopping criterion. The tables below detail the total training epochs required for convergence, and the absolute wall-clock training times in seconds.

Table 5: Epochs to convergence comparison on the facility location problem (50 seeds)

| Method | Architecture | Epochs to convergence | | | |
|--------|-------------|-----------------------|---|---|---|
| | | 10 instances | 20 instances | 100 instances | 200 instances |
| PFL | Shared Model | 67.94 ± 2.95 | 46.12 ± 2.00 | 19.22 ± 0.75 | 10.90 ± 0.29 |
| | Separate Models | 100.70 ± 4.74 | 92.30 ± 4.29 | 35.78 ± 1.15 | 17.12 ± 0.37 |
| | Separate Heads | 59.96 ± 3.27 | 45.10 ± 2.40 | 20.40 ± 0.79 | 14.26 ± 0.40 |
| | Shared + FiLM | 35.96 ± 3.35 | 29.32 ± 1.93 | 18.04 ± 0.90 | 12.12 ± 0.64 |
| DFL | Shared Model | 65.56 ± 2.76 | 43.80 ± 1.80 | 15.64 ± 0.47 | 11.28 ± 0.31 |
| | Separate Models | 54.24 ± 1.50 | 52.32 ± 1.28 | 27.20 ± 0.67 | 17.60 ± 0.63 |
| | Separate Heads | 58.48 ± 2.09 | 42.24 ± 1.13 | 18.64 ± 0.55 | 14.22 ± 0.38 |
| | Shared + FiLM | 51.70 ± 3.34 | 39.76 ± 1.83 | 19.56 ± 0.72 | 15.40 ± 0.68 |
| DFL with data augmentation | Shared Model | 16.34 ± 0.43 | 11.64 ± 0.36 | 8.02 ± 0.43 | 8.56 ± 0.40 |
| | Separate Models | 28.66 ± 0.83 | 17.64 ± 0.40 | 9.50 ± 0.33 | 8.40 ± 0.37 |
| | Separate Heads | 19.54 ± 0.54 | 14.92 ± 0.44 | 9.98 ± 0.36 | 9.22 ± 0.38 |
| | Shared + FiLM | 18.16 ± 0.60 | 15.36 ± 0.59 | 10.86 ± 0.42 | 9.82 ± 0.42 |

# F   Results for Joint Model

In this section, we present an empirical evaluation of a "Joint Model" baseline, where the parallel feature vectors of all items are concatenated into a single large input vector to predict the full parameter vector simultaneously. We showcase its performance relative to the shared and FiLM-modulated architectures. The following results showcase the limitation of this approach. Because each individual parameter only correlates with its own specific features, a large multi-output model processing the entire concatenated vector is forced to map features to completely uncorrelated outputs. The model struggles to filter out and ignore the irrelevant feature vectors for each distinct output, leading to severe data inefficiency and poor overall performance.

Table 6: Wall-clock training time (s) comparison on the facility location problem (50 seeds)

| Method | Architecture | Wall-clock training time (s) | | | |
|---|---|---|---|---|---|
| | | 10 instances | 20 instances | 100 instances | 200 instances |
| PFL | Shared Model | $12.15 \pm 2.20$ | $2.48 \pm 0.12$ | $1.97 \pm 0.08$ | $2.77 \pm 0.43$ |
| | Separate Models | $35.71 \pm 4.15$ | $21.43 \pm 1.14$ | $15.77 \pm 0.54$ | $14.15 \pm 0.99$ |
| | Separate Heads | $17.67 \pm 2.12$ | $10.89 \pm 0.71$ | $10.38 \pm 0.47$ | $13.48 \pm 0.88$ |
| | Shared + FiLM | $9.48 \pm 1.89$ | $3.28 \pm 0.22$ | $5.14 \pm 0.28$ | $7.21 \pm 0.72$ |
| DFL | Shared Model | $62.83 \pm 3.80$ | $71.05 \pm 3.55$ | $116.28 \pm 5.60$ | $165.78 \pm 8.35$ |
| | Separate Models | $61.98 \pm 3.35$ | $96.84 \pm 3.42$ | $226.12 \pm 9.33$ | $297.14 \pm 16.54$ |
| | Separate Heads | $69.26 \pm 5.00$ | $79.02 \pm 3.71$ | $153.21 \pm 8.41$ | $241.01 \pm 10.10$ |
| | Shared + FiLM | $55.84 \pm 4.19$ | $72.85 \pm 4.66$ | $165.71 \pm 8.46$ | $270.69 \pm 15.94$ |
| DFL with data augmentation | Shared Model | $135.00 \pm 4.50$ | $191.06 \pm 7.03$ | $643.49 \pm 37.38$ | $1379.40 \pm 60.52$ |
| | Separate Models | $253.52 \pm 12.05$ | $300.48 \pm 10.25$ | $800.19 \pm 40.81$ | $1425.85 \pm 76.20$ |
| | Separate Heads | $174.60 \pm 7.53$ | $262.41 \pm 11.76$ | $820.23 \pm 42.88$ | $1465.80 \pm 88.87$ |
| | Shared + FiLM | $170.59 \pm 8.69$ | $274.06 \pm 12.84$ | $862.48 \pm 53.89$ | $1573.70 \pm 110.23$ |

Table 7: Epochs to convergence comparison on the knapsack problem (50 seeds)

| Method | Architecture | Homogeneous | Heterogeneous | |
|---|---|---|---|---|
| | | | Small range | Large range |
| PFL | Shared Model | $15.00 \pm 0.43$ | $15.00 \pm 0.52$ | $15.10 \pm 0.66$ |
| | Separate Models | $28.30 \pm 0.62$ | $25.60 \pm 0.75$ | $22.70 \pm 0.63$ |
| | Separate Heads | $20.40 \pm 0.91$ | $18.80 \pm 0.69$ | $18.30 \pm 0.68$ |
| | Shared + FiLM | $20.70 \pm 0.59$ | $18.80 \pm 0.78$ | $17.80 \pm 0.66$ |
| DFL | Shared Model | $15.30 \pm 0.63$ | $12.50 \pm 0.46$ | $11.60 \pm 0.33$ |
| | Separate Models | $19.80 \pm 0.57$ | $22.10 \pm 0.61$ | $19.90 \pm 0.65$ |
| | Separate Heads | $20.00 \pm 0.74$ | $19.30 \pm 0.85$ | $20.60 \pm 0.95$ |
| | Shared + FiLM | $17.70 \pm 0.70$ | $17.60 \pm 0.77$ | $17.80 \pm 0.82$ |
| DFL with data augmentation | Shared Model | $11.70 \pm 0.37$ | $11.50 \pm 0.36$ | $12.10 \pm 0.41$ |
| | Separate Models | $13.30 \pm 0.55$ | $14.50 \pm 0.69$ | $13.50 \pm 0.67$ |
| | Separate Heads | $12.50 \pm 0.54$ | $13.90 \pm 0.69$ | $14.60 \pm 0.51$ |
| | Shared + FiLM | $11.80 \pm 0.45$ | $11.90 \pm 0.40$ | $12.10 \pm 0.41$ |

Table 8: Wall-clock training time (s) comparison on the knapsack problem (50 seeds)

| Method | Architecture | Homogeneous | Heterogeneous | |
|---|---|---|---|---|
| | | | Small range | Large range |
| PFL | Shared Model | $1.09 \pm 0.04$ | $1.04 \pm 0.04$ | $1.24 \pm 0.12$ |
| | Separate Models | $16.79 \pm 0.50$ | $13.93 \pm 0.54$ | $13.24 \pm 0.45$ |
| | Separate Heads | $7.43 \pm 0.34$ | $6.59 \pm 0.29$ | $6.56 \pm 0.27$ |
| | Shared + FiLM | $3.51 \pm 0.11$ | $3.14 \pm 0.12$ | $3.43 \pm 0.23$ |
| DFL | Shared Model | $12.31 \pm 0.53$ | $69.57 \pm 2.96$ | $52.30 \pm 1.80$ |
| | Separate Models | $25.86 \pm 0.75$ | $101.42 \pm 3.11$ | $74.96 \pm 3.16$ |
| | Separate Heads | $21.88 \pm 0.80$ | $95.86 \pm 5.07$ | $84.22 \pm 4.56$ |
| | Shared + FiLM | $17.15 \pm 0.70$ | $92.97 \pm 5.33$ | $71.98 \pm 3.49$ |
| DFL with data augmentation | Shared Model | $92.24 \pm 2.88$ | $717.75 \pm 20.79$ | $597.74 \pm 20.29$ |
| | Separate Models | $155.71 \pm 7.37$ | $638.31 \pm 35.97$ | $484.32 \pm 25.48$ |
| | Separate Heads | $114.26 \pm 5.30$ | $650.65 \pm 34.02$ | $567.90 \pm 26.04$ |
| | Shared + FiLM | $100.67 \pm 4.32$ | $665.45 \pm 34.34$ | $488.17 \pm 21.39$ |

Table 9: Facility Location Relative Regret (%) comparing the JointModel architecture against Shared and Shared+FiLM DFL.

| Architecture | Relative regret (%) | | | |
|---|---|---|---|---|
| | **10 instances** | **20 instances** | **100 instances** | **200 instances** |
| Shared DFL | $2.95 \pm 0.22$ | $2.80 \pm 0.21$ | $2.71 \pm 0.18$ | $2.67 \pm 0.18$ |
| Shared + FiLM DFL | $6.68 \pm 1.54$ | $4.08 \pm 1.25$ | $1.74 \pm 0.12$ | $1.68 \pm 0.12$ |
| JointModel DFL | $56.76 \pm 1.53$ | $54.36 \pm 1.44$ | $43.11 \pm 1.08$ | $33.42 \pm 0.83$ |

Table 10: Knapsack Relative Regret (%) comparing the JointModel architecture against Shared and Shared+FiLM DFL.

| Architecture | Relative regret (%) | | |
|---|---|---|---|
| | **Homogeneous** | **Heterogeneous** | |
| | | **Small range** | **Large range** |
| Shared DFL | $4.43 \pm 0.04$ | $4.04 \pm 0.04$ | $3.13 \pm 0.03$ |
| Shared + FiLM DFL | $4.42 \pm 0.04$ | $3.98 \pm 0.04$ | $2.96 \pm 0.03$ |
| JointModel DFL | $22.55 \pm 0.14$ | $22.62 \pm 0.19$ | $22.66 \pm 0.26$ |

Table 11: Newsvendor Absolute Regret comparing the JointModel architecture against Shared and Shared+FiLM DFL (Seed 3 dropped in 50 parameters).

| Architecture | Absolute regret | | |
|---|---|---|---|
| | **10 parameters** | **30 parameters** | **50 parameters** |
| Shared DFL | $70.67 \pm 2.15$ | $218.54 \pm 8.67$ | $347.09 \pm 14.08$ |
| Shared + FiLM DFL | $61.61 \pm 2.10$ | $176.65 \pm 2.69$ | $310.71 \pm 14.02$ |
| JointModel DFL | $94.58 \pm 0.83$ | $899.12 \pm 23.32$ | $2541.99 \pm 11.77$ |

