# OpenReview forum: "Accounting for Heterogeneous Parameters in Decision-Focused Learning"
_TMLR — Under review for TMLR_

### Review · Reviewer_qpqG · 2026-04-15

**Summary Of Contributions:**

This paper studies the problem of decision focused learning (DFL), and focuses on the presence of variables/inputs of same nature, that are heterogeneous in the sense they do not play a symmetric role in the problem at hand. The paper provides the following contributions:
- mathematical definition and characterization of homegeneous/heterogeneous variables
- proposition of 3 methods to train models, and potentially predicting heterogeneous variables with separate models
- they provide experiments on synthetic data, as well as some (partially, see below) real data, to evaluate their different methods and show improvement over the PFL and single model DFL baseline

**Audience:**

Yes

**Audience Explanation:**

The machine learning community has recently grown some interest towards the problem of DFL (in opposition to predict-then-optimize or prediction focused learning). In consequence, I believe this work is aligned with TMLR audience.

**Claims And Evidence:**

No

**Claims Explanation:**

The paper is well written, the mathemetical definitions are well introduced. Beyond these definitions, the mathematical results are not specifically difficult: they give a precise characterization of homogeneity, which motivates the methodological approaches introduced in Section 5.

I however find the main motivation of this work slightly dubious. This work is mainly motivated by the claim that "conventionally, one would train a single model that is shared across all parameters". However, I am not sure of such a claim. Intuitively, I would have rather trained a model that takes as input the features of all the items as a single large vector (e.g., $[x_1, \ldots, x_n]$ where $x_i$ are the features of the item i) aiming at predicting all the parameters (e.g., the vector $[y_1,\ldots, y_n]$). I can understand this is not the common practice, but I would need citations supporting that, as my first intuition would have been to do as I describe here. Note that this way of doing could be a fourth benchmark to compare with in your experimental section.

**Requested Changes:**

1) do you have any references supporting that "conventionally, one would train a single model that is shared across all parameters"?

2) could you add in the experimental section the benchmark I mentioned above?

3) Currently, the paper mostly adopts the point of view that *parameters are mostly heterogeneous in practice*, which leads to methods that proceed as if all these parameters were indeed heterogeneous. However, I think one could still leverage some homogeneity in the problem parameters, as we could use shared predictors/weights for them. This could yield to a reduction in the number of parameters of the models to train, hence requiring less data. I would thus like that this work does not see their theoretical results as negative, but also as some nice way to detect homegenous parameters, to leverage this structure.

4) Following my point 3), we could even go further and make the notion of homogeneity/heterogeneity something that is quantifiable and continuous, in the sense that we could somehow quantify that two parameters are "nearly homogeneous". This would allow to make this  weight sharing idea even stronger, as it could go beyond "exact homogeneity".

5) Section 6.2.2 shows experiments on California Housing prices. I am however slightly disappointed, as here only the variables $(x_i,y_i)$ are from real data: the knapsack weights $z_i$ and budget $W$ are synthetic. I think the DFL litterature is currently missing an illustration on real data, with a real task; and I would have enjoyed to find such a thing here. Also, the dataset is not very clearly described here. A knapsack problem is given by 40 items from the dataset, but how are these 40 items chosen? what do they represent? It is hard to know here if they are really of the same nature (ie having same distribution), whether there is some correlation between these items, etc. I would have liked a more detailed explanation of the experimental setup and the dataset here.
-------------------

# Minor remarks

- second paragraph after Definition 1 (and maybe elsewhere too): referring to homogeneous and heterogeneous variables is a bit confusing, as it is not an absolute property (with this definition), but only a relative one between two variables.
- Lemma 1: what if the argmin are not unique? Is this an equality between sets then?
- The involution $\tau$ introduced in Lemma 2 is a bit mysterious at the beginning. Maybe it would be nice to actually describe at that point the involution that we will be considering, which is given in the proof of Lemma 3.
- Regarding the involution of Lemma 3, do we really need the "there exists an involution $\tau$..." statement? It seems that you precisely know what is the involution (given in the proof), which would make the lemma more precise (and again, this involution less mysterious)
- After the proof of Lemma 3, I would have liked more details explaining why "even if there is an involution $\tau$ such that [...], this does not necessarily imply heterogeneity"
-  the data augmentation scheme is only valid if $(x_i,y_i)$ have same distribution across different items (more precisely, that these random variables are exchangeable). Some disclaimer about that might be helpful for further reuse of the proposed methods.
- how many training instances are used for Table 2?

---

> ### Author Response · Authors · 2026-06-15
> **Rebuttal (Part 1/2)**
>
> We thank the reviewer for the thoughtful and constructive feedback. We appreciate the positive assessment of the paper's motivation and theoretical formalization, as well as the concrete suggestions for improvement.
>
> **Is it really conventional to use a single model shared across all parameters?**
>
> Yes. This observation was one of the motivations for the present work. As discussed in the manuscript, the dominant paradigm in the DFL literature is to train a single predictive model that is shared across all parameters, with parameter predictions obtained through repeated invocations of that model on the corresponding feature vectors (Shah et al., 2022; Shah et al., 2024; Wilder et al., 2019; Mandi et al., 2022; Mulamba et al., 2021; Mandi & Guns, 2020; Ferber et al., 2020). To the best of our knowledge, the only exception is the synthetic benchmark introduced by Elmachtoub & Grigas (2022) and later adopted by PyEPO (Tang et al., 2024), where separate models are used because the parameters are generated from distinct underlying mappings.
>
> **What about a model that jointly predicts all parameters from a concatenated feature vector?**
>
> We agree that a joint model predicting all parameters simultaneously from a concatenated feature representation is a natural alternative architecture. In fact, we considered this approach during the development of the paper. However, preliminary experiments consistently showed substantially weaker performance than the architectures reported in the manuscript, and we therefore chose not to include it in the original submission. Following the reviewer's suggestion, we added a dedicated appendix section reporting these results. Conceptually, the difficulty is that each parameter depends only on its own associated feature vector. A large multi-output model operating on the fully concatenated input must therefore learn to map the relevant features to the relevant outputs while simultaneously identifying and ignoring the many irrelevant feature vectors for each prediction. In practice, this leads to reduced predictive performance, which is reflected in the experimental results.
>
> **Theoretical results could also be used to identify homogeneous parameters and share weights. This motivates a quantifiable and continuous measure of heterogeneity.**
>
> We strongly agree and thank the reviewer for this valuable suggestion. In the revised manuscript, we introduce a quantifiable "degree of heterogeneity" metric. This is a continuous metric ($d_\sigma$) that can be used to quantify the expected violation of regret invariance between parameter pairs under a given error scale $\sigma$. This measure can serve as a diagnostic tool to evaluate parameter heterogeneity, and allows practitioners to identify when parameters are 'nearly homogeneous', to then share weights across the corresponding predictive components. We also use this measure in our revised experimental section to connect our experiments more directly to our theory. We specifically use it to evaluate the effect of varying the degree of heterogeneity on the performance of the different architectures.
>
> **The California Housing setup is not sufficiently described. How are the 40 items chosen? Are they really of the same nature? Is there correlation between items?**
>
> We agree that the original description was too brief. In our setup, we first randomly shuffle the California Housing dataset and then partition the districts into disjoint training, validation, and test sets. Each optimization instance is subsequently generated by sampling 40 districts from the appropriate split, with each district representing a single item in the knapsack problem. Because the districts are shuffled before the train/validation/test partitioning and instances are generated through random sampling, the items within a knapsack instance do not carry any meaningful spatial or temporal correlations. Consequently, the items can reasonably be viewed as being drawn from the same underlying distribution. We have expanded the description of the benchmark in the revised manuscript to make these design choices explicit.
>
> **The California Housing benchmark is only partially real, since the optimization problem is synthetic**
>
> We agree with this observation. Unfortunately, this is largely representative of the current state of DFL benchmarking, where real predictive datasets are often combined with synthetic optimization problems. In fact, many DFL methods are evaluated primarily on *fully* synthetic benchmarks, where both the predictive and optimization components are artificially generated. We therefore view the California Housing benchmark as a useful intermediate setting that introduces realistic predictive data, rather than as a fully real-world decision-making task. We agree that more fully integrated real-world DFL benchmarks would be highly valuable for the community.

---

> > ### Author Response · Authors · 2026-06-15
> > **Rebuttal (Part 2/2)**
> >
> > **What if the argmin is not unique? Is this an equality between sets then?**
> >
> > We thank the reviewer for highlighting this issue. We have added a brief discussion of the non-unique solution case. While the main exposition uses notation that assumes a unique optimal solution for readability, our theoretical characterizations naturally extend to optimization problems with multiple optimal solutions through a set-valued formulation. We now clarify this explicitly in the manuscript. However, for notational simplicity and readability, we forego a fully set-valued exposition to avoid unnecessary technical complexity that does not alter our core findings.
> >
> > **The involution $\tau$ is somewhat mysterious when first introduced.**
> >
> > We agree that the original exposition made $\tau$ appear somewhat abstract. To improve readability, we added intuitive explanations of Proposition 2 and restructured the proof. We now make the role of $\tau$ clearer by interpreting it as the symmetry that compensates for a parameter transposition while preserving relative solution quality.
> >
> > **More details would be helpful regarding why the weaker involution conditions do not imply homogeneity.**
> >
> > Intuitively, this is because homogeneity requires the symmetry condition to hold across the entire supported solution space $Z^\star$, not merely at the optimum. We have added a brief discussion of this to the revised manuscript, and touch on this point in much more detail in Appendix B.
> >
> > **The data augmentation scheme is only valid if the different parameters follow the same distribution. Some disclaimer about that might be helpful for further reuse of the proposed methods.**
> >
> > We agree with this observation. In the revised manuscript, we made this assumption explicit. More specifically, for the augmented instances generated by parameter permutations to remain strictly in-distribution, it is not sufficient to assume that all parameters depend on their associated features through the same predictive mapping. One must additionally assume that the parameter-feature pairs are identically distributed (or, equivalently in this context, exchangeable). We now state this assumption explicitly. We also note, however, that this identical-distribution assumption is also implicitly made whenever a single predictive model is shared across all parameters, which is the conventional choice in existing work.
> >
> > **How many training instances are used for Table 2?**
> >
> > We realize this information was easy to miss in the original manuscript. The experimental setup is described in full detail in Appendix C: all training sets contained 250 optimization instances (corresponding to 40×250=10,000 parameter-feature pairs), while the validation and test sets each contained 129 optimization instances. To improve clarity, we now make this information more explicit in the main text of the revised manuscript.
> >
> > We thank the reviewer again for the constructive feedback and hope that the revisions and clarifications described above address the concerns raised in the review.

---

### Review · Reviewer_8Ttm · 2026-05-18

**Summary Of Contributions:**

This paper considers the predict-then-optimize setting, where there are two main paradigms: (1) prediction focus learning (PFL) where the main goal is to train a model that minimizes the difference between the predicted optimal value (or solution) and the ground truth optimal value, and (2) decision focus learning (DFL), where the goal is instead to maximize the performance of the prediction model with respect to some downstream task. Under this problem setting, the papers focus on the latter case and argue that the architecture used in DFL should properly handle the heterogeneity of the decision variables. Based on that observation, the authors propose two things: (1) architecture-wise: they propose to use (a) positional encoding, (b) separate models, or (c) the same backbone but different heads to predict different decision values, and (2) data augmentation. Finally, the authors show that their proposal leads to better results in some datasets.

**Audience:**

Yes

**Audience Explanation:**

The paper is in the decision-focused learning, which is quite a hot topic recently.

**Claims And Evidence:**

No

**Claims Explanation:**

I feel that the contents/claims in this paper are not rigorous enough. See the requested changes section for more information.

**Requested Changes:**

I think that the paper is not well-written. I have many concerns throughout this papers:


### 1. In the Introduction section

1. The final sentence in the second paragraph in the Introduction section seems off a bit. Please consider rewording to improve the flow there.

2. In the second paragraph in the Introduction section, the final sentence is also vague and needs more elaboration.

3. In the second and third paragraphs (about PFL and DFL), we also need more elaboration. Are the authors talking about predicting the optimal solution vs. predicting the optimal objective value? I mean, the wording makes the audience feel that way, but obviously, it does not. Besides, what do you mean by “downstream decision quality”? What is the downstream task here? Why does predicting the optimal solution relate to downstream decision quality? How do models trained with DFL generally lead to better decisions, and in what way? We need to be more precise here.

4. The wording in the whole fourth paragraph in the introduction section is also very odd, disconnected, and not carefully elaborated. Besides, some of the word choices are very odd (e.g., “separate invocation”). Please consider rewriting the whole paragraph.

5. Again, the next sections (the premise of the contribution of this work, and the example on knapsack) are very unclear to me. I think the author needs to move the example to the beginning of the introduction, to give the audience a taste of what PFL and DFL are with this concrete example.

Overall, I believe that this introduction is not well-written and requires a massive revamp to make it publishable.

### 2. In the Background section

1. In Eq. (1): argmin is typically a set, so we cannot write $z^\star(y) = \arg \min_{z}$ without extra premises (like clarifications, notation overloading, or some assumptions on the given minimization problems).

2.  What do you mean by “correlated with known feature …. according to some distribution $P$”? And later on, you say: “each parameter … depends on its features … to the same underlying mapping”? Are you equating the distribution with a mapping? What is $\\mathbb{R}^f$ in $x_i \\in mathbb{R}^{f}$? Why $x$ have the same dimensionality $n$ as $y$?

The Related Work section seems a bit disconnected; I'm not sure why the authors put it after the background section.

### 3. In the Heterogeneous Parameter section
1. Though Lemmas 1, 2, and Proposition 3 are straightforward, I think the authors should have a discussion before or right after stating that those results. Give the author a taste of what they are about (let’s say, in plain English), and why they are important on your analysis.

2. Again, everything in this paper operates under the assumption that $z^\\star(y)$ is a single vector (the solution set is unique, which originates from Eq. 1). This one is NOT a standard assumption, does not hold true in all most every case. If this is not satisfied, I am afraid that the whole idea of this section, or even worse, the whole paper, is invalid. So, it is better to discuss this very, very carefully.

3. There is a typo in the second last paragraph: $… f(z^\\star(y); y) = f(z^\\star(\\sigma_{ij}(y), \\sigma_{ij})$, it should be $… f(z^\\star(y); y) = f(z^\\star(\\sigma_{ij}(y), \\sigma_{ij}(y))$

Overall, I think the theoretical contribution here is a bit minimal. So I will try to judge the contribution of this paper purely on the empirical aspect, which is in the later sections.

### 4. On the Accounting for heterogeneous parameters section

1. The architectural adaptations proposed in Section 5.1 (positional encodings, independent models, shared backbones with specific heads) are highly standard paradigms in the broader machine learning literature (e.g., multi-task learning). While applying them to DFL to combat parameter heterogeneity is a sound empirical exercise, the authors must be careful not to overstate the architectural novelty of this work. I strongly recommend revising the introduction and Section 5 to explicitly clarify that the core contribution is conceptual/diagnostic (formalizing parameter heterogeneity in DFL) and that the architectures are simply applications of standard off-the-shelf ML techniques used to resolve this newly identified issue.”

2. In Section 5.2, the authors claim to 'propose a data augmentation scheme'. While I acknowledge the authors' clarification that permuting parameters within an instance generates structurally novel optimization problems (and differs from standard batch shuffling), permutation-based augmentation itself is a highly standard technique in machine learning. I suggest softening the novelty claim here. Rather than stating 'we propose this scheme', it would be more accurate to frame this as 'adapting' or 'applying' standard permutation augmentation to synthesize novel optimization instances for DFL.

### 5. On the Experiments section

1. While the authors adequately detail the network dimensions, batch sizes, and epochs in Appendix C, the paper completely omits actual wall-clock computational training times. The authors themselves note in Section 3 that DFL suffers from scalability issues due to repeated solver calls. Introducing parameter-specific models or heads naturally raises concerns about computational overhead. The authors must include a discussion and an empirical table comparing the training and inference times of the proposed architectures against the shared PFL/DFL baselines.


## Final comment
All in all, I feel that the presentation of this paper needs to be significantly improved. Besides, it is very hard for me at the current state to evaluate the key contribution of this paper. I will wait for other reviews as well as discussing with the AE to have a better assessment on this draft.

---

> ### Author Response · Authors · 2026-06-15
> **Rebuttal (Part 1/2)**
>
> We thank the reviewer for the detailed and constructive feedback. We appreciate the careful assessment of the manuscript and the provided suggestions. Below, we address each comment in turn and describe the corresponding revisions made to the paper.
>
> **Concerns about the introduction, problem setting and overall presentation being unclear**
>
> We have substantially restructured the introduction to improve its narrative flow and clarity. To clarify the problem setting here: in the predict-then-optimize problem, the aim is not to predict the optimal solution or the optimal objective value of the optimization problem. Instead, the machine learning model is strictly predicting the unknown parameters (typically objective coefficients) of an optimization problem from contextual features. The distinction between prediction-focused learning (PFL) and decision-focused learning (DFL) lies in how this predictive model is trained.  PFL solely aims to make parameter predictions as accurately as possible, and ignores the optimization problem during the training loop. Instead, DFL actually solves the optimization problem using the predicted parameters in each forward pass during training, allowing the model to prioritize minimizing precisely those parameter prediction errors that would cause the solver to make highly suboptimal decisions. This generally leads to significantly improved decision-making. To clarify common terminology in this field: the "downstream task" is the optimization problem itself (e.g., finding the shortest path over predicted arc costs on a graph), and "downstream decision quality" refers to how well the produced solution performs under the actual, ground-truth parameters. We have extensively revised the Introduction section to make the paper more welcoming to readers not yet fully familiar with the predict-then-optimize problem setting, and to ensure precision regarding what is being predicted versus what is being optimized.
>
> **On the existence of non-unique solutions**
>
> We thank the reviewer for highlighting this issue. We have added a brief discussion of the non-unique solution case. While the main exposition uses notation that assumes a unique optimal solution for readability, our theoretical characterizations naturally extend to optimization problems with multiple optimal solutions through a set-valued formulation. We now clarify this explicitly in the revised manuscript. However, for notational simplicity and readability, we forego a fully set-valued exposition to avoid unnecessary technical complexity that does not alter our core findings.
>
> **What do you mean by “correlated with known feature …. according to some distribution”?**
>
> The intended meaning is the standard supervised-learning setting: each optimization problem parameter is associated with contextual features, and training instances are sampled from an underlying joint distribution over features and parameters.
>
> **Are we equating the distribution with a mapping?**
>
> This is a good observation. In the original manuscript, our wording in the data augmentation section was indeed too loose. The predictive mapping and the data-generating distribution are distinct concepts, and we have clarified this in the revised manuscript. More specifically, for the augmented instances generated by parameter permutations to remain strictly in-distribution, it is not sufficient to assume that all parameters depend on their associated features through the same predictive mapping. One must additionally assume that the parameter-feature pairs are identically distributed. We now state this assumption explicitly. We note, however, that this identical-distribution assumption is also implicitly made whenever a single predictive model is shared across all parameters, which is the conventional choice in existing work.
>
> **What is $\mathbb{R}^f$ in $x_i \in \mathbb{R}^f$? Why does $x$ have the same dimensionality $n$ as $y$?**
>
> Here, $f$ denotes the dimensionality of an individual feature vector. The vectors $x$ and $y$ both have $n$ elements because each optimization parameter $y_i$ ​ is associated with its own feature vector $x_i$. Thus, $x=(x_1, …, x_n)$ represents a collection of $n$ feature vectors, where each $x_i \in \mathbb{R}^f$ provides the contextual information used to predict the corresponding parameter $y_i$​ .
>
> **The Related Work section seems a bit disconnected; I'm not sure why the authors put it after the background section**
>
> We have revised the transitions into and out of the related work section to improve its placement within the overall paper.

---

> > ### Author Response · Authors · 2026-06-15
> > **Rebuttal (Part 2/2)**
> >
> > **Intuitive interpretations of theoretical results**
> >
> > We added intuitive plain-English interpretations for the notion of supported solutions, Lemma 1, and Proposition 2, providing a clearer interpretation of the results and their role in motivating the remainder of the paper. We have also removed the original standalone Lemma 2 and incorporated its contents directly into the proof of Proposition 2 using a “claim/proof of claim” structure. In retrospect, it served purely as a technical stepping stone rather than a standalone theoretical contribution. Integrating it improved readability of the overall section.
> >
> > **There is a typo in the second last paragraph**
> >
> > Thank you for pointing this out. We have corrected the typo in the revised manuscript.
> >
> > **Overall, I think the theoretical contribution here is a bit minimal. So I will try to judge the contribution of this paper purely on the empirical aspect, which is in the later sections.**
> >
> > The theoretical contribution of the original manuscript is the formalization of parameter heterogeneity through regret invariance, and the characterization of the conditions under which two parameters can be considered homogeneous. In particular, Proposition 2 shows that parameter homogeneity is equivalent to the existence of a symmetry of the supported solution space that preserves solution quality under parameter transposition. This phenomenon has, to the best of our knowledge, not previously been studied in the predict-then-optimize literature.
> >
> > Additionally, in the revision we extended the analysis beyond the original binary homogeneous/heterogeneous distinction by introducing a continuous degree of heterogeneity metric, ($d_\sigma$), which quantifies the expected violation of regret invariance between parameter pairs. This provides a more nuanced characterization of heterogeneous parameters and serves as a practical diagnostic tool.
> >
> > **The architectural adaptations proposed in Section 5.1 (positional encodings, independent models, shared backbones with specific heads) are highly standard paradigms in the broader machine learning literature (e.g., multi-task learning)**
> >
> > We want to highlight that the goal of this work is not the introduction of fundamentally new neural architectures. To better reflect this, we have revised the framing throughout the manuscript accordingly. We now more clearly position the primary contribution of the paper as conceptual and diagnostic rather than architectural: namely, formalizing parameter heterogeneity in predict-then-optimize problems and demonstrating its impact on DFL. The architectural adaptations are presented as simple and practical mechanisms for accounting for this phenomenon rather than as novel neural architectures. In addition, we replaced the original parameter-index encoding approach with a more expressive FiLM-based conditioning mechanism that leverages structural problem attributes.
> >
> > **The theoretical and empirical sections appear disconnected**
> >
> > We have addressed this concern in the revision by introducing a continuous degree of heterogeneity metric and by using it in a dedicated experiment. The results show that, as the degree of heterogeneity increases, parameter-aware architectures increasingly outperform the fully shared model, providing direct empirical support for our theoretical analysis.
> >
> > **The authors must include a discussion and an empirical table comparing the training and inference times of the proposed architectures against the shared PFL/DFL baselines**
> >
> > To address this concern, we introduced an early stopping criterion (rather than a fixed number of epochs), and added a dedicated appendix reporting both the epochs until convergence and wall-clock training times for all methods. The results show that accounting for parameter heterogeneity does not slow training in practice. The parameter-aware architectures converge in a comparable number of epochs and exhibit similar wall-clock training times to the fully shared model, indicating that the observed performance gains are not achieved at the expense of slower convergence.
> >
> > We thank the reviewer again for the careful assessment and constructive feedback. We hope that the revisions and clarifications outlined above address the raised concerns.

---

### Review · Reviewer_6q5f · 2026-05-29

**Summary Of Contributions:**

This paper studies a new setting of the predict-then-optimize problem: different parameters influence decision quality in distinct ways, leading to varying impacts of prediction errors across parameters. The authors formally define this setting, introduce the concept of “heterogeneous parameters” and propose three prediction models capable of learning parameter-specific mappings.

**Additional Comments:**

Have you considered heterogeneous parameters with correlations? Can the proposed methods still work in such a scenario?

**Audience:**

Yes

**Audience Explanation:**

The focus of the paper and the observations on the effects of “heterogeneous parameters” are interesting and useful.

**Claims And Evidence:**

No

**Claims Explanation:**

1. The theoretical explanation and analysis of heterogeneous parameters are quite limited. Detailed interpretations of the lemmas and propositions would be beneficial, and the properties of heterogeneous parameters should be explored further.

2. The theoretical part and the experimental part seem separate and not clearly connected.

3. The proposed models are somewhat straightforward. Encoding the parameter index as part of the features can only be regarded as a baseline, and using a separate parameter-specific model or a shared backbone with parameter-specific heads are common approaches when predicting parameters separately.

**Requested Changes:**

What are the training times for the three proposed methods? Would training time be a bottleneck when using separate parameter-specific models for large-scale problems? Reporting the training time of the proposed methods would be beneficial.

---

> ### Author Response · Authors · 2026-06-15
> **Rebuttal**
>
> We thank the reviewer for their time and constructive feedback. Below, we address each comment in turn and describe the corresponding revisions made to the manuscript.
>
> **The theoretical explanation and analysis of heterogeneous parameters are quite limited. Detailed interpretations of the lemmas and propositions would be beneficial, and the properties of heterogeneous parameters should be explored further.**
>
> We agree that the original presentation could be made more accessible. In the revised manuscript, we substantially restructured the theoretical section. We added intuitive, plain-English explanations for the key theoretical results, including the notion of supported solutions, Lemma 1, and Proposition 2. Furthermore, we extended the analysis beyond the original binary homogeneous/heterogeneous distinction by introducing a continuous degree of heterogeneity metric, ($d_\sigma$), which quantifies the expected violation of regret invariance between parameter pairs. This provides a more nuanced characterization of heterogeneous parameters and serves as a practical diagnostic tool.
>
> **The theoretical part and the experimental part seem separate and not clearly connected.**
>
> We have addressed this concern in the revision by introducing a continuous degree of heterogeneity metric and using it in a dedicated experiment. The results show that, as the degree of heterogeneity increases, parameter-aware architectures increasingly outperform the fully shared model, providing direct empirical support for our theoretical analysis.
>
> **The proposed models are somewhat straightforward. Encoding the parameter index as part of the features can only be regarded as a baseline, and using a separate parameter-specific model or a shared backbone with parameter-specific heads are common approaches.**
>
> We want to highlight that the goal of this work is not the introduction of fundamentally new neural architectures. To better reflect this, we revised the framing throughout the manuscript and explicitly position our contribution as conceptual and diagnostic rather than architectural. Our goal is to formalize and diagnose parameter heterogeneity in predict-then-optimize problems and to demonstrate that accounting for it in simple ways can substantially improve decision quality. In addition, we replaced the original one-hot parameter-index conditioning approach with a more expressive FiLM-based conditioning mechanism that modulates the predictions based on structural problem attributes, providing a stronger parameter-aware architecture.
>
> **What are the training times for the three proposed methods? Would training time be a bottleneck when using separate parameter-specific models for large-scale problems?**
>
> To address this concern, we introduced early stopping and added a dedicated appendix reporting both the epochs until convergence and wall-clock training times for all methods. The results show that accounting for parameter heterogeneity does not slow training in practice. The parameter-aware architectures converge in a comparable number of epochs and exhibit similar wall-clock training times to the fully shared model, indicating that the observed performance gains are not achieved at the expense of slower convergence.
>
> **Have you considered heterogeneous parameters with correlations? Can the proposed methods still work in such a scenario?**
>
> The approach we propose does not assume that all parameters must be treated independently. In fact, the newly introduced degree of heterogeneity metric can be used to identify parameter pairs that are nearly homogeneous, for which weight sharing may be beneficial. In practice, one may opt for architectures that selectively share weights between subsets of parameters rather than adopting a fully shared or fully separate architecture. We have therefore expanded the discussion of future work to include adaptive parameter clustering based on the degree of heterogeneity, allowing groups of similar parameters to be learned jointly while preserving parameter-specific modeling where needed.
>
> We thank the reviewer again for the constructive feedback and hope that the revisions and clarifications described above address the concerns raised in the review.

---

### Author Response · Authors · 2026-06-15
**Summary of Changes (Part 1/2)**

We thank the reviewers for their constructive and insightful feedback. Based on your comments, we have significantly revised our manuscript. We believe these modifications have clarified the paper's core message, and have addressed the concerns expressed by the reviewers.
In the revised manuscript, the implemented changes are highlighted in red for ease of tracking. Below, we provide a structured overview of the major updates, followed by detailed point-by-point responses to each reviewer's specific comments.

## Abstract, Introduction, and Overall Framing

**Restructured introduction:** We have overhauled the introduction to improve the narrative flow and provide greater clarity. (Reviewer 8Ttm)

**Motivating example:** We have strengthened the beginning of the introduction with a knapsack example to clarify the idea behind predict-then-optimize, and which directly connects with Example 1 and Example 2 given later on in the paper. (Reviewer 8Ttm)

**Clarifying the predictive task:** We have further clarified the predict-then-optimize problem setup, to clear up that the machine learning model is strictly predicting the *unknown parameters* of an optimization problem from contextual features, rather than predicting the optimal solution or the optimal objective value of the optimization problem. (Reviewer 8Ttm)

**Reframing of conceptual vs. architectural novelty:** We adjusted the phrasing throughout the manuscript to clarify that introducing intricate novel architectures is not the objective of this work. Instead, our goal is conceptual and diagnostic: we formalize the fact that parameters in predict-then-optimize problems can respond differently to prediction errors, and demonstrate that accounting for this via simple adaptations significantly improves downstream decision quality. (Reviewer 6q5f and Reviewer 8Ttm)

## Background and related work:

**Addressing non-unique solution sets:** We added an explicit clarification regarding the uniqueness of the optimal solution. While our formalization uses notation that assumes a single optimal solution for clarity and readability, our theoretical characterizations generalize to problems with multiple optimal solutions via a set-based formulation. However, we forego a fully set-valued exposition to avoid unnecessary technical complexity that does not alter our core findings. (Reviewer 8Ttm)

**Smoother transitions:** We added transitional text to the end of Section 2 to flow more naturally into the related work section. (Reviewer 8Ttm)

**Connecting to related work:** We revised the closing paragraph of Section 3 to clarify the link between prior literature that primarily focuses on devising learning signals or improving scalability, and our distinct focus on parameter heterogeneity. (Reviewer 8Ttm)

## Heterogeneous parameters:

**Restructuring:** We revised the structure of this section. The existing theoretical derivations were grouped into a new subsection titled “conditions for homogeneity”, and a separate subsection with a newly introduced “degree of heterogeneity” has been added.

**Intuitive theoretical explanations:** We added clearer explanations to our theoretical statements to give readers a plain-English intuition of the concept of supported solutions, Lemma 1, and Proposition 2. (Reviewer 6q5f, Reviewer 8Ttm)

**Removal of standalone Lemma 2:** We removed the original standalone Lemma 2 and incorporated its contents directly into the proof of Proposition 2 using a “claim/proof of claim” structure. In retrospect, it served purely as a technical stepping stone rather than a standalone theoretical contribution. Integrating it improved readability of the overall section. (Reviewer 6q5f and Reviewer 8Ttm)

**Continuous “degree of heterogeneity" metric:** To move beyond a binary homogeneous/heterogeneous distinction, we introduced a quantifiable "degree of heterogeneity" metric: This is a continuous metric ($d_\sigma$) that can be used to quantify the expected violation of regret invariance between parameter pairs under a given error scale $\sigma$. This can serve as a diagnostic tool to evaluate parameter heterogeneity, and allows practitioners to identify when parameters are 'nearly homogeneous', to then share weights across the corresponding predictive components. We also use this measure in our revised experimental section to connect our experiments more directly to our theory. We specifically use it to evaluate the effect of varying the degree of heterogeneity on the performance of the different architectures. (Reviewer qpqG and Reviewer 6q5f)

---

> ### Author Response · Authors · 2026-06-15
> **Summary Of Changes (Part 2/2)**
>
> ## Accounting for heterogeneous parameters:
>
> **Introduction of FiLM conditioning:** We replaced the naive one-hot encoded parameter index approach with a more sophisticated Feature-wise Linear Modulation (FiLM) conditioning layer, in which an auxiliary network $g_{\phi}$ scales and shifts base predictions based on known, structural problem attributes (such as constraint coefficients). We also revised Figure 2 to reflect the replacement of the one-hot parameter index architecture with the FiLM conditioning architecture. (Reviewer 6q5f)  This architecture has the added benefit that parameters with similar structural attributes (and thus a lower expected degree of heterogeneity) are automatically biased by the architecture to be treated more similarly. (Reviewer qpqG)
>
> **Offline data augmentation:** We restructured the data augmentation scheme (Algorithm 1) from an online per-iteration permutation loop to an offline data augmentation scheme executed prior to training. This adjustment led to a more controllable amount of augmentation, and stabilized training. We also added an explanation on the two ways in which this augmentation strategy can benefit DFL under heterogeneous parameters. Finally, we explicitly mention the identical-distribution assumption of the data augmentation scheme.
>
>
> ## Experiments:
>
> **Empirical evaluation using the degree of heterogeneity:** We used the newly introduced continuous heterogeneity metric ($d_\sigma$) to systematically verify that as the degree of heterogeneity increases, the performance gap between the parameter-aware architectures and the fully shared model widens. (Reviewer 6q5f and Reviewer qpqG)
>
> **Experiments with FiLM:** In the benchmark descriptions, we added a description of the structural attributes given to the FiLM model $g_\phi$ for each benchmark. We also added a more detailed description of the FiLM conditioning network to Appendix C. We included the new Shared + FiLM architecture in all experimental results across the benchmarks (Tables 1, 2, and 3). The empirical results demonstrate that this is a competitive approach that reliably outperforms the conventional shared model in the presence of parameter heterogeneity. (Reviewer 6q5f and Reviewer 8Ttm)
>
> **Reporting of epochs until convergence and wall-clock times:** We adjusted the training from using a fixed epoch count to an early-stopping criterion (stopping when validation regret fails to improve by $\ge 1\%$ for 5 epochs). This allowed us to address reviewer concerns regarding the scalability of parameter-specific heads and models by tracking and appending convergence time statistics to the newly added Appendix E. We report both the exact number of epochs to convergence and wall-clock training times. (Reviewer 6q5f and 8Ttm)
>
> **Joint model baseline evaluation:** To address questions regarding an alternative multi-output network architecture, we added a dedicated appendix section evaluating a "Joint Model" baseline (concatenating all parallel item feature vectors into a single input vector). We demonstrate empirically, and discuss conceptually, why this approach does not perform well and was thus left out of the original manuscript. (Reviewer qpqG)
>
> ## Conclusion
>
> **Overall update:** We updated the conclusion to reflect the changes discussed above.
>
> **Expanded future work:** We broadened our future work section to include a discussion of adaptive parameter clustering using the degree of heterogeneity metric.
>
>
> We apologize for the slight delay in providing this response. The beginning of the rebuttal period overlapped with a recent conference, limiting our availability during the first week. We would like to sincerely thank the action editor for graciously granting an extension to the discussion period to compensate for this overlap.